# Catch bond models may explain how force amplifies TCR signaling and antigen discrimination

Hyun-Kyu Choi [1,2], Peiwen Cong [1,2], Chenghao Ge[1,2,11], Aswin Natarajan[3,4], Baoyu Liu[1,2,12], Yong Zhang[5,6,7], Kaitao Li [1,2], Muaz Nik Rushdi [1,2,13], Wei Chen [8,9], Jizhong Lou [5,6,7], Michelle Krogsgaard[3,4] & Cheng Zhu [1,2,10] ✉

The TCR integrates forces in its triggering process upon interaction with pMHC. Force elicits TCR catch-slip bonds with strong pMHCs but slip-only bonds with weak pMHCs. We develop two models and apply them to analyze 55 datasets, demonstrating the models' ability to quantitatively integrate and classify a broad range of bond behaviors and biological activities. Comparing to a generic two-state model, our models can distinguish class I from class II MHCs and correlate their structural parameters with the TCR/pMHC's potency to trigger T cell activation. The models are tested by mutagenesis using an MHC and a TCR mutated to alter conformation changes. The extensive comparisons between theory and experiment provide model validation and testable hypothesis regarding specific conformational changes that control bond profiles, thereby suggesting structural mechanisms for the inner workings of the TCR mechanosensing machinery and plausible explanations of why and how force may amplify TCR signaling and antigen discrimination.

Antigen recognition via interactions between the T-cell antigen receptor (TCR) and peptide major histocompatibility complex (pMHC) molecule is essential for T-cell activation, differentiation, proliferation, and function[1]. Mechanical forces applied to αβTCR via engaged pMHC substantially increase antigen sensitivity and amplify antigen discrimination[2–9]. As a fundamental force-elicited characteristic, strong cognate pMHCs form catch-slip bonds with TCR where bond lifetimes increase with force until reaching a peak, and decrease as

force increases further, whereas weak agonist and antagonist pMHCs form slip-only bonds with TCR where bond lifetimes decrease monotonically with increasing force[2–4,6–10]. However, the mechanism underlying the correlation between the force-lifetime pattern and the ability for force on TCR to induce T-cell signaling remains unclear.

An intuitive hypothesis is that catch bonds prolong TCR engagement with pMHC, which allows the process of CD3 signal initiation to proceed a sufficient number of phosphorylation steps to the threshold

[1]Wallace H. Coulter Department of Biomedical Engineering, Georgia Institute of Technology and Emory University, Atlanta, GA 30332, USA. [2]Parker H. Petit Institute for Bioengineering and Biosciences, Georgia Institute of Technology, Atlanta, GA 30332, USA. [3]Laura and Isaac Perlmutter Cancer Center, New York University Grossman School of Medicine, New York, NY 10016, USA. [4]Department of Pathology, New York University Grossman School of Medicine, New York, NY 10016, USA. [5]National Laboratory of Biomacromolecules, Institute of Biophysics, Chinese Academy of Sciences, Beijing 100101, China. [6]Key Laboratory of RNA Biology, CAS Center for Excellence in Biomacromolecules, Institute of Biophysics, Chinese Academy of Sciences, Beijing 100101, China. [7]University of the Chinese Academy of Sciences, Beijing 100049, China. [8]Department of Cell Biology, Zhejiang University School of Medicine, Hangzhou 310058, China. [9]Department of Cardiology of the Second Affiliated Hospital, Zhejiang University School of Medicine, Hangzhou 310058, China. [10]George W. Woodruff School of Mechanical Engineering, Georgia Institute of Technology, Atlanta, GA 30332, USA. [11]Present address: Amgen Inc., One Amgen Center Dr., Thousand Oaks, CA 91320, USA. [12]Present address: Department of Pathology, University of Utah School of Medicine, Salt Lake City, UT 84112, USA. [13]Present address: Medtronic CO., Minneapolis, MN 55432, USA. ✉e-mail: cheng.zhu@bme.gatech.edu

for downstream signal propagation, as proposed by the kinetic proofreading model[11]. However, this hypothesis faces two challenges upon scrutiny of multiple TCR–pMHC systems. First, since bond lifetime vs force profiles are monotonically decreasing for slip-only bonds but bell-shaped for catch-slip bonds, the local bond type would depend on the force range, which may be catch bonds in one force regime but slip bonds in another force regime. Second, TCR–pMHC interactions exhibiting catch-slip bonds often have longest lifetimes around 10–20 pN (refs. 2–4,6–10.) and it has been reported that upon engaging pMHC, T cells would exert 12–19 pN endogenous forces on the TCR in a signaling-dependent fashion[12]. However, the relevance of this force range to T-cell signaling remains incompletely understood. More perplexingly, some signal-inducing pMHCs form catch-slip bonds with TCRs but exhibit shorter lifetime than other pMHCs that do not induce signaling by, and form slip-only bonds with, the same TCRs even in the optimal force range[13]. These observations prompt the questions of what mechanism underlies the association of TCR–pMHC bond type with the T-cell signaling capacity and what impact the 10–20 pN force range has on TCR mechanotransduction. To answer these questions, requires an in-depth analysis of the multiple datasets with mathematical models, which was lacking.

Slip and catch bonds refer to two opposite effects of physical force on biomolecular interactions: increasing or decreasing their off-rate of dissociation, respectively[14,15]. Because force tends to be disruptive and destabilizing, slip bonds are intuitive, whereas catch bonds are counter-intuitive. Since excessive force can rupture even covalent bonds[16], continued force increase will eventually overpower any catch bond, turning it to a slip bond after an "optimal" force where the off-rate is minimal[2–4,6–9,15]. Slip bond is commonly modeled by the Bell

equation[17], which assumes the off-rate $k$ of a molecular bond dissociating along a single pathway in a one-dimensional (1D), single-well energy landscape to be an exponential function of force, $k(F) = k_0 e^{\frac{\delta_0^* F}{k_B T}}$. Here, $k_0$ is the transition rate at zero force, $F$ is tensile force, $\delta_0^*$ is the force-free distance from the bound state at the bottom of the energy well to the top of the energy barrier known as the "transition state", $k_B$ is the Boltzmann constant, and $T$ absolute temperature[17]. Several models have been developed to account for catch-slip bond behavior. Most introduced two dissociation pathways and/or two bound states in a two-dimensional (2D) energy landscape that is tilted by force[18] (Fig. 1a). One noticeable exception is that of Guo et al. where dissociation is modeled to start from a single bound state along a single pathway through a 1D energy landscape based on the physical process of peeling a polymer strand with force until the transition state is reached[19]. A distinct advantage of this model is its ability to relate the force-induced deformation of the energy landscape to the force-induced conformational change of the molecular complex (which all other models lack), thereby connecting parameters of the abstract energy landscape to the structural-elastic properties of the interacting molecules in question.

Besides binding properties of the TCR–pMHC complex, its structural features and conformational changes have been suggested to be important for TCR triggering. For example, TCR–pMHC docking orientation has been correlated to its ability to trigger T-cell signaling[13,20,21]. Partial unfolding or allosteric regulation of either the TCR and/or MHC molecules has been inferred from mechanical experiments and steered molecular dynamics (SMD) simulations of pulling single TCR–pMHC bonds[4,6,22,23], which have been supported by mutagenesis experiment[6]. Whereas the extent of these conformational

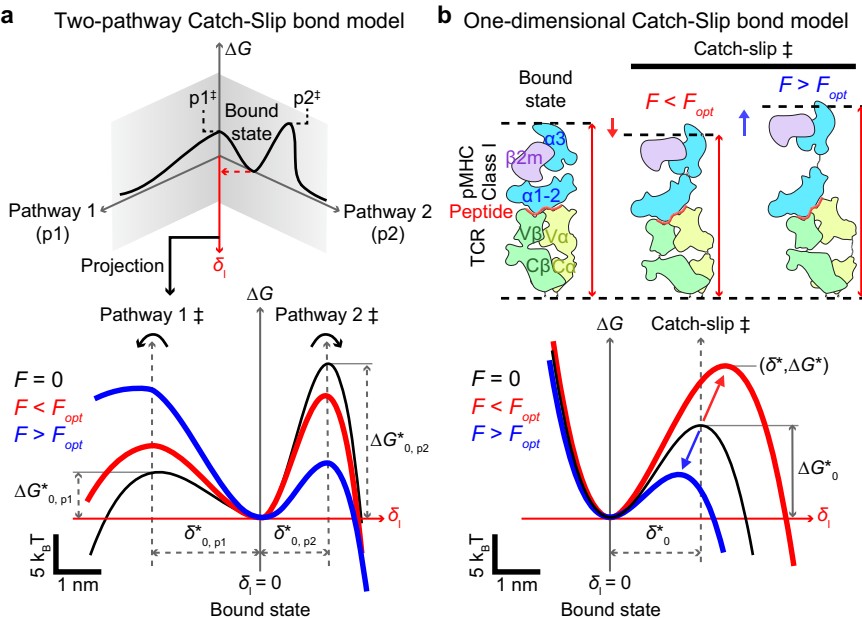

**Fig. 1 | Comparison between the two-pathway model and the TCR–pMHC-I model for catch bond. a** Upper: The 2D energy landscape of the two-pathway model where the bond is trapped in the bound-state energy well by two energy barriers that resist dissociation along two pathways, p1 and p2, with two distinct transition states, p1‡ and p2‡. The application of force projects the energy landscape towards a dissociation pathway along force ($\delta_l$). Lower: Force also tilts the energy landscape, raising the energy barrier of the first pathway and lowering the energy barrier of the second pathway on the energy landscape projection (red and blue) relative to their positions in the absence of force (black). **b** Lower: The proposed 1D energy landscape of TCR–pMHC-I model with a single transition state $\delta$. Below an optimal value ($F_{opt}$), force raises the energy barrier (*red*) relative to the zero-force conformation (*black*) by contraction of flexible regions due to entropic

fluctuation. Above $F_{opt}$, force lowers the energy barrier (blue) by stretching the molecular complex. Together, these two mechanisms give rise to a catch-slip bond. Upper: Schematics of the TCR–pMHC-I structure (left) and its conformational changes that correspond to low (middle) and high (right) forces. Note that in the lower panels of both (**a**, **b**), the energy wells in the absence and presence of force are aligned to the same level and the energy barrier levels are allowed to change in response to force. This convention is made throughout this paper for clear visualization because, as far as kinetic rate theory is concerned, only the energy difference between the energy barrier and energy well matters. However, this convention does not mean to suggest that force can only change the energy barrier level but not the energy well level in a real protein complex structure; to the contrary, both are possible[70].

changes has been correlated to the level of TCR–pMHC catch bonds[4,6,22], the two have not been integrated into a mathematical model to explore their potential connection.

Here we develop two such models, one for each MHC class, to describe both αβTCR catch-slip and slip-only bonds. The model development follows Kramers' kinetic rate theory and uses polymer physics models to construct a 1D energy landscape for single-state, single-pathway dissociation that incorporates the structures, elastic properties, and force-induced conformational changes of the TCR–pMHC-I/II complexes at the sub-molecular level, which includes domain stretching, hinge rotation, and molecular extension. Incorporating the force-induced conformational changes into the energy landscape formulation allows force to shift the energy barrier up in low forces and down in high forces, thereby giving rise to catch-slip bonds along a single-dissociation pathway in a 1D energy landscape (Fig. 1b). We applied our models and a published model[24] to analyze 49 TCR–pMHC bond lifetime vs force datasets published to date in 9 papers by four laboratories measured using biomembrane force probe (BFP)[2,3,6,8,10,13,25,26] or optical tweezers (OT)[4], which have spatial, temporal, and force resolutions in the order of nanometer, sub-millisecond, and piconewton for BFP[27–29] and better for OT[30]. Six additional datasets were generated in two sets of experiments using specific mutations at remote regions away from the binding interface to change force-induced conformational changes in the TCR or MHC to test our models. The total datasets include 12 TCRs and their mutants expressed on the cell membrane or coated on beads interacting with corresponding panels of both classes of pMHCs without coreceptor engagement. This analysis demonstrate our models' structural and physical parameters to quantitatively integrate and classify a broad range of catch-slip and slip-only bond behaviors as well as their corresponding biological activities. Our models were rigorously validated by extensively comparing theory with experiment, testing the model assumptions and predictions, and using mutagenesis to alter specific conformational changes in the TCR–pMHC structure under force to modulate the catch-bond profiles. By constructing the energy landscape underlying our models and investigating its properties, we obtain mechanistic insights into the inner workings of the TCR–pMHC mechanosensory machinery. By examining the correlation of the model parameters with the biological activities of a large number of TCR–pMHC-I/II systems, we explain how force-elicited catch bond may amplify TCR signaling and antigen discrimination.

## Results

### Model development

**Model goal.** Kramers' kinetic rate theory treats bond dissociation as state transition in a 1D energy landscape $\triangle G^*(\delta_l)$ from a free-energy well (bound state) over a barrier (transition state) along the dissociation coordinate $\delta_l$[31]. Following ref. 19 to adapt the linear-cubic model of Dudko et al.[32] but allow force $F$ to deform the original energy landscape by an amount of $-\delta_l\gamma(F)$, the energy landscape takes the form of

$$\triangle G(\delta_l,F) = \frac{3\triangle G_0^*}{2}\left(\frac{\delta_l}{\delta_0^*} - \frac{1}{2}\right) - 2\triangle G_0^*\left(\frac{\delta_l}{\delta_0^*} - \frac{1}{2}\right)^3 - \delta_l\gamma(F) \quad (1)$$

where $\delta_0^*$ and $\triangle G_0^*$ are the differences in dissociation coordinates and free-energy levels, respectively, between the transition state and bound state of the original force-free energy landscape. The corresponding force-dependent kinetic rate is

$$k(F) = k_0\sqrt{1 - \frac{2\delta_0^*\gamma(F)}{3\triangle G_0^*}}\exp\left(\frac{\triangle G_0^*}{k_BT}\left(1 - \left(1 - \frac{2\delta_0^*\gamma(F)}{3\triangle G_0^*}\right)^{3/2}\right)\right) \quad (2)$$

where $k_0$ is the dissociation rate at zero force[19]. Letting $\gamma \sim F$ recovers from Eq. (2) the Dudko–Hummer–Szabo (DHS) model[32], and further assuming $|2\delta_0^*\gamma(F)/(3\triangle G_0^*)| \ll 1$ reduces it to the Bell model[17]. The condition for $k$ to be able to model catch-slip bond is the derivative $k'(F_0) = 0$ where $F_0 > 0$. This translates to two conditions: the barrier height $\triangle G^* = \triangle G_0^*\left(1 - \frac{2\delta_0^*\gamma(F_0)}{3\triangle G_0^*}\right)^{3/2} = k_BT/3$ or $\gamma'(F_0) = 0$. The first condition requires the energy change $-\delta_l\gamma$ induced by $F_0$ to lower the energy barrier height to $k_BT/3$ located at $\delta_0^*\left(\frac{k_BT}{2\triangle G_0^*}\right)^{1/3}$. The second condition requires $\gamma$ to be a biphasic function of $F$. This excludes the Bell model and the DHS model because both of their $\gamma$ functions depend on $F$ monotonically; as such, only allow force to tilt the energy landscape for the energy difference between the energy barrier and energy well to change monotonically, i.e., either slip or catch bond. Our goal is to construct a biphasic $\gamma(F)$ with appropriate structural-elastic dependency to account for TCR–pMHC conformational changes from the bound state to the transition state, as analyzed by the BFP and OT experiments where a tensile force is applied to its two ends to modulate bond dissociation[2–4,6–10], which will allow force to deform the energy landscape in such a way that the energy difference between the energy barrier and energy well would increase at low forces but decrease at large forces as required by catch-slip bond (Fig. 1b), i.e., catch-slip bond.

**Key model assumptions.** Following the reasoning of Guo et al.[19], $\delta_l\gamma(F) = \int_0^F \delta_z(f)df$ where the integrand $\delta_z(f) = z(f) - z_0(f)$ is the projection on the force direction of the change induced by force $f$ of the TCR–pMHC extension at the transition state relative to its extension at the bound state. For $\gamma$ to depend on $F$ biphasically as required for describing catch-slip bonds, $\delta_z$ should be a biphasic function of $f$ as discussed later. Therefore, dissociation occurs because the system moves in the energy landscape along the dissociation coordinate $\delta_l$ from the bound state to the transition state by a distance $\delta^* = \delta_0^*\left(1 - \frac{2\delta_0^*\gamma(F)}{3\triangle G_0^*}\right)^{1/2}$ (ref. 19). We assume that the differential contour length along the force-transmission path across the TCR–pMHC structure (i.e., summing up all contour lengths of various domains connected at nodes of force action, as depicted in red lines in Fig. 2a, for MHC-I) at the transition state $l$ and bound state $l_0$ can serve as a dissociation coordinate, i.e., $\delta_l = l - l_0$. $\delta_z$ is the projection of $\delta_l$ on the z axis—the direction of the pulling force (Fig. 2a). When only contour lengths are considered, $\delta_l = \delta_0^*$, which serves as a criterion for finding the best-fit parameters (Fig. 2a, Bound state and Supplementary Model Derivations, Eq. (11)).

Suggested by single-molecule OT[4], BFP[6] and magnetic tweezers[6] experiments as well as steered MD (SMD) simulations[6], we assume that force-induced TCR–pMHC dissociation is accompanied by conformational changes in the TCR, MHC, or both. Specifically, we assume that at the bound state, force induces elastic extension of the TCR–pMHC structure as a whole (Fig. 2a, bound state); but as the system moves toward the transition state for dissociation, conformational changes may occur, which may include disruption of intramolecular interfaces, hinge rotation, and partial unfolding of interdomain joints (Fig. 2a, transition state). To include appropriate details of these proposed conformational changes at the sub-molecular level into the expression of $\delta_z$, we model the TCR–pMHC structure as a system of semi-rigid bodies representing the whole complex as well as various globular domains connected by semi-flexible polymers that allow extension and hinge rotation under mechanical loads (Fig. 2a, transition state). Specifically, we assume that force may induce disruption of the MHC α1α2–β2m interdomain bond, thereby shifting the mechanical load originally borne by this bond to the α1α2–α3 joint to induce its partial unfolding, as observed in SMD simulations[6]. As such, the MHC α3

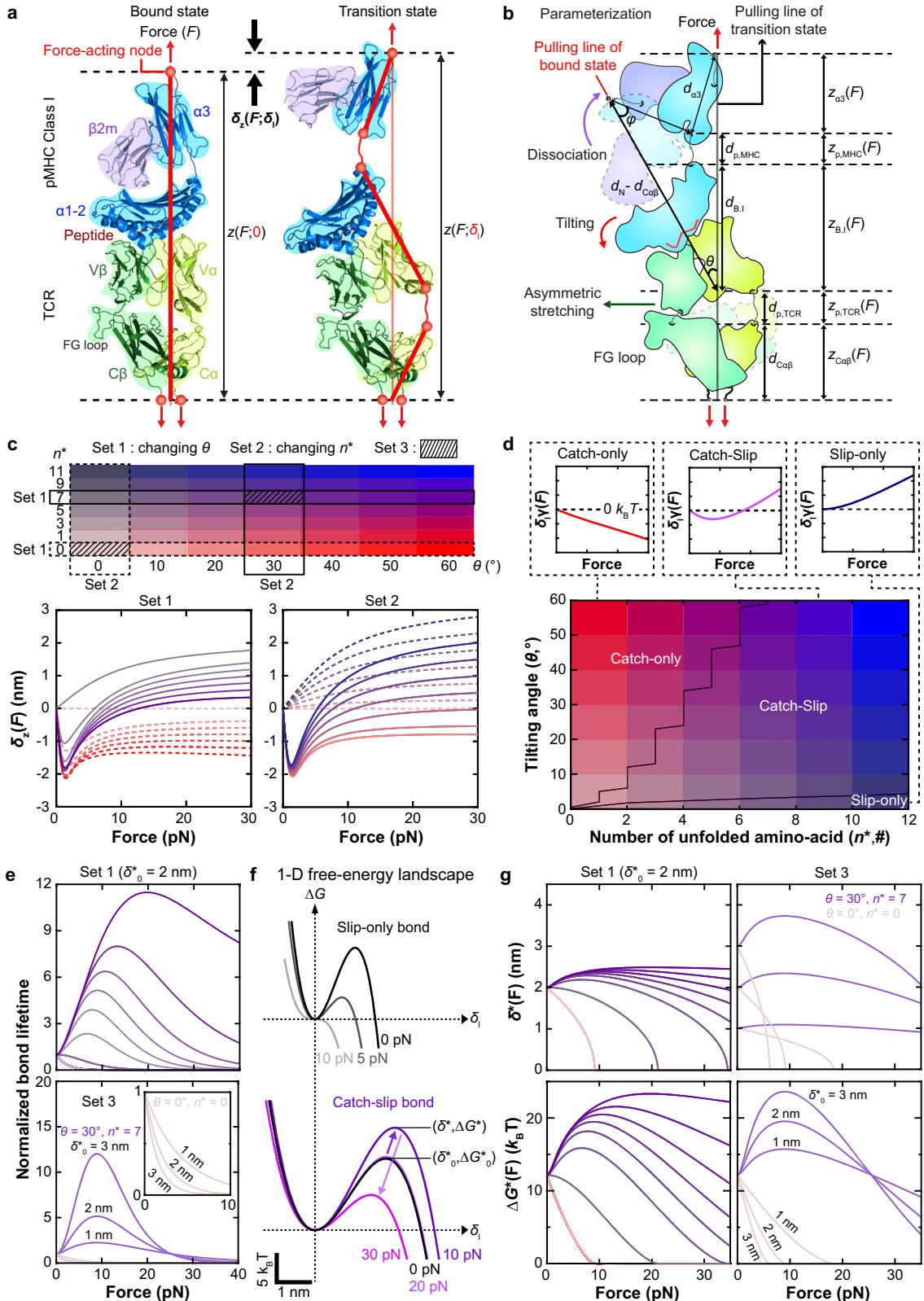

domain would change its length and rotate about its C-terminus (Fig. 2b). Since the TCRβ subunit has also been proposed to undergo FG-loop-regulated conformational change[4,22,23], we assume that disruption of the $\alpha_1\alpha_2$–$\beta_{2m}$ joint would result in tilting of the TCR–pMHC bonding interface and shifting of the mechanical load from the TCR Vβ-Cβ joint to the Vα-Cα joint, leading to partial unfolding of the Vα-Cα joint (Fig. 2b). This increased stretching of the Vα-Cα joint

relative to the Vβ–Cβ joint is assumed to result from strengthening of the Vβ–Cβ joint by the FG-loop[5]. At the transition state, therefore, we treat the MHC $\alpha_3$ domain ($d_{\alpha_3}$), the MHC $\alpha_1\alpha_2$ domains bound to the TCR Vαβ domains ($d_{B.I}$), and the TCR Cαβ domains ($d_{C\alpha\beta}$) as three semi-rigid bodies connected by two unfolded peptide chains of the MHC $\alpha_1\alpha_2$–$\alpha_3$ joint ($d_{p,MHC}$) and the TCR Vα–Cα joint ($d_{p,TCR}$) (Fig. 2b). At the bound state, neither disruption of intramolecular bonds nor

**Fig. 2 | Structure, mechanics, and characteristics of the TCR–pMHC-I catch-bond model. a** Force-induced conformational changes of a TCR–pMHC-I complex as it traverses from the bound state (left) to the transition state (right). The diagrams of the 2C TCR α (yellow) β (green) subunits and the DEVA peptide (red) bound to the H2-K$^b$ (various domains indicated) are based on snapshots from SMD simulations performed on the complex structure (2CKB) at the initial time (bound state) and a later time (transition state)[6]. The force-transmission path is shown as red lines connecting the force-acting nodes. **b** Various contributions to the total extension projected on the force axis: rotation of the α$_3$–β$_{2m}$ domains about the MHC C-terminus upon dissociation of the β$_{2m}$–α$_1$α$_2$ interdomain bond ($z_{\alpha3}$), relative rotation between α$_3$ and α$_1$α$_2$ about their stretched interdomain hinge ($z_{p,MHC}$), tilting of the MHC α$_1$α$_2$ complexed with TCR Vαβ ($z_{B.I}$), rotation about and extension of the Vα–Cα interdomain hinge ($z_{p,TCR}$), and extension of the Cαβ and rotation about their C-termini ($z_{C\alpha\beta}$). Two α$_3$-β$_{2m}$ structures are shown: before (light colors) and after (dark colors) β$_{2m}$ dissociation from α$_1$α$_2$, with two parameters describing their contributions to the total extension: = the distance between the α3 C- and N-termini excluding the α$_1$α$_2$-α3 hinge and θ = the angle between the normal direction of the TCR–pMHC bonding interface at the bound state (cf. (**a**), left) and the tilted direction at the transition state (cf. (**a**), right). **c** Extension change vs force curves (lower) for the color-matched $n^*$ and θ values (upper). The left panel (set 1 in upper table) shows the effect of changing θ with ($n^* = 7$) and without ($n^* = 0$) partial unfolding. The right panel (set 2) shows the effect of changing $n^*$ with (θ = 30°) and without (θ = 0°) tilting. **d** $n^*$-θ phase diagram showing three parameter domains: slip-only, catch-slip, and catch-only respectively colored by red-purple, purple-blue, and blue-black. Upper insets indicate corresponding energy change $\delta_l\gamma$ vs force curves for each bond type. **e** Theoretical normalized bond lifetime vs force curves for indicated parameters. The upper and lower panels show the respective effects of changing θ and $\delta_0^*$ from the set 1 and set 3 parameters defined in (**c**), respectively. **f** Energy landscapes expressed as families $\triangle G$ vs $\delta_l$ curves for a range of forces for slip-only (upper) and catch-slip (lower) bonds. The bound state is located at the origin $\triangle G = 0$ and $\delta_l = 0$ in the absence and presence of force by the convention stated in Fig. 1 legends. The transition state has an energy of $\triangle G^*$ located at $\delta^*$ when $F > 0$ and $\Delta G_0^*$ located at $\delta_0^*$ when $F = 0$. **g** Plots of transition-state location $\delta^*$ (upper) and height of energy barrier $\triangle G^*$ (lower) vs force $F$ for changing θ while keeping $\delta_0^*$ constant (left) or changing $\delta_0^*$ while keeping θ constant (right) for the indicated values from parameter table in (**c**). Source data are provided as a Source Data file.

partial unfolding of interdomain joints occurs, as mentioned earlier, allowing the whole TCR–pMHC ectodomain (ECD) complex to be modeled as one semi-rigid body ($d_N$).

**Force-induced energy change.** To derive an expression for the last term on the right-hand side of Eq. (1), we model the semi-rigid bodies $d_i$ ($i$ = N, α$_3$, B.I., and Cαβ to, respectively, denote the whole TCR–pMHC ECD structure as well as its indicated domains) as three-dimensional freely-jointed chains (FJC) and employ polymer physics to obtain their force-extended length $d_i(f)$ from their force-free length $d_{i,c}$ (ref. 33) (Supplementary Model Derivations, Eq. (5)).

The assumed partial unfolding of the α$_1$α$_2$–α$_3$ joint and the Vα–Cα joint are based on suggestions from single-molecule OT[4], BFP[6] and magnetic tweezers[6] experiments as well as SMD simulations[6]. We model these unstructured polypeptides as extensible worm-like chains (eWLC) and employ polymer physics to obtain their force-induced extension $d_{p,i}(f)$ ($i$ = MHC and TCR) from their force-free, folded state, which has zero length[34] (Supplementary Model Derivations, Eq. (7)).

Upon projecting the various force-induced extensions described above onto the force axis, we obtain $z$ components of five contributions to the TCR–pMHC length increase at the transition state: extension of the MHC α$_3$ domain ($z_{\alpha3}$), unfolding of the MHC α$_1$α$_2$–α$_3$ interdomain joint ($z_{p,MHC}$), extension of bonding interface that includes the MHC α$_1$α$_2$ domains bound to the TCR Vαβ domains ($z_{B.I}$), unfolding of the Vα–Cα joint ($z_{p,TCR}$), and extension of the TCR Cαβ domains ($z_{C\alpha\beta}$) (Fig. 2b). Finally, we obtain:

$$\delta_l\gamma(F) = \int_0^F \left[ z_{\alpha3}(f) + z_{p,MHC}(f) + z_{B.I}(f) + z_{p,TCR}(f) + z_{C\alpha\beta}(f) - z_N(f) \right] df \quad (3)$$

### Model characterization

**Model constants and parameters.** The FJC model constants for the 1st, 3rd, 5th, and 6th terms in the integrand on the right-hand side of Eq. (3) include the force-free lengths $d_{i,c}$ and the elastic modulus of the folded globular domains $E_c$, all available from the literature. The 2nd and 4th terms are proportional the respective numbers of amino acids in the polypeptides of the partially unfolded MHC α$_1$α$_2$–α$_3$ joint ($n_{p,MHC}$) and TCR Vα–Cα joint ($n_{p,TCR}$), which can be combined as the product of the total unfolded amino acid number $n^* = n_{p,MHC} + n_{p,TCR}$, the average contour length per unfolded amino acid $l_c$, and the extension per unit contour length $z_{u,p}(f)$. The eWLC model constants for $z_{u,p}(f)$ include the average persistence length per unfolded amino acid $l_p$ and the elastic modulus of polypeptides $E_p$ (Supplementary Table 1).

After applying model constraints and the approximation $\triangle G_0^* \sim \ln(k_w/k_0)$ where $k_w \sim 10^6 \text{ s}^{-1}$ is known as the prefactor

(Supplementary Model Derivations), the model parameters are reduced to five: three structural parameters ($d_{\alpha3}$, θ, $n^*$) and two biophysical parameters ($k_0$, $\delta_0^*$), for describing dissociation of TCR–pMHC-I bonds. We will determine these parameters by comparing the model predictions with experimental measurements, and in doing so, illustrate the ability of our model to use a relatively low number of parameters to capture the coarse-grained structure and conformational changes at the sub-molecular level during TCR–pMHC-I dissociation.

**Model features and properties.** To explore the general features and properties of the model, we plotted $\delta_z$ vs $F$ for two $n^*$ values and a range of θ values as well as two θ values and a range of $n^*$ values (Fig. 2c). Conceptually, force-heightened energy barrier (relative to the aligned energy well, not to the force-free energy landscape) generates catch bonds and force-lowered energy barrier (again relative to the aligned energy well) produces slip bonds (Fig. 2d, catch-only and slip-only). Since $-\delta_l\gamma$ represents the energy input by force $F$ to the original energy landscape, a biphasic $\delta_l\gamma$, (i.e., at some forces energy is added into, and at other forces energy is released from, the energy landscape) is required to create catch-slip bonds (Fig. 2d, catch-slip); correspondingly, $\delta_z$ is required to have a root at positive $F$ where catch-bond transitions to slip bond (Fig. 2c). The parameter domains capable of generating catch, catch-slip, and slip bonds are mapped on an $n^*$-θ phase diagram (Fig. 2d), showing that our model can describe catch-slip bond if and only if $n^* > 0$, $θ > 0$, and $\delta_l\gamma(\infty) > 0$ (Fig. 2d, catch-slip). Conformationally speaking, catch-slip bonds require partial unfolding of the MHC α$_1$α$_2$–α$_3$ and/or TCR Vα–Cα joints and tilting of the TCR–pMHC bonding interface, a prediction consistent with previous results of SMD simulations and single-molecule experiments[6].

For single-bond dissociation from a single bound state along a single pathway, the reciprocal dissociation rate should be equal to the average bond lifetime. Regardless of the bond type, the reciprocal zero-force off-rate controls the y-intercept of the bond lifetime vs force curves. We plotted the theoretical bond lifetime (normalized by its zero-force value) $k_0/k$ vs force $F$ for a range of $n^*$, θ, and $\delta_0^*$ to examine how the model parameters control the bond lifetime vs force profile (Fig. 2e). Consistent with Fig. 2c, d, only if $n^* > 0$ and $θ > 0$ can our model describe catch-slip bond. Increasing the tilting angle θ results in more pronounced catch-slip bonds with longer lifetimes that peak at higher forces (Fig. 2e, set 1 with $\delta_0^* = 2$ nm). By comparison, increasing $\delta_0^*$ changes the level of slip-only bonds if $n^* = 0$ and θ = 0, but prolongs lifetime of catch-slip bonds (until cross-over at a higher force) without changing the force where lifetime peaks if $n^* > 0$ and $θ > 0$ (Fig. 2e, set 3).

To understand physically how our model describes catch-slip bonds, we plotted the energy landscape using Eq. 1 (Fig. 2f). Setting $\theta = 0$ generates a family of $\triangle G$ vs $\delta_1$ curves where the energy barrier is suppressed monotonically with increasing force, indicating a slip-only bond (Fig. 2f, slip-only bond). By comparison, setting $\theta > 0$ results in a family of $\triangle G$ vs $\delta_1$ curves where increasing force initially raises (i.e., adding energy to the system), then lowers (i.e., releasing energy from the system), the relative energy barrier height, indicating a catch-slip bond (Fig. 2f, catch-slip bond, also see Fig. 1b). We also examine how the transition-state location (Fig. 2g, $\delta^*(F)$) and energy barrier height (Fig. 2g, $\triangle G^*(F)$) change with force for a range of $\theta$ and $\delta_0^*$ values that give rise to slip-only bonds and catch-slip bonds. Noticeably, at fixed $\theta$ values, both rates by which the transition-state location and the energy barrier height change with force are accelerated by increasing $\delta_0^*$ (Fig. 2g, set 3), suggesting that this parameter can be used as a measure for force sensitivity. Interestingly, increasing $\theta$ slows the decrease in both the transition-state location and energy barrier height with force at higher values, suggesting that the tilting angle controls the range at which force sensitivity can last (Fig. 2g, set 2 with $\delta_0^* = 2$ nm).

## Model validation

**Model's capability to fit data.** To test our model's validity, we used it to analyze 9 class I-restricted TCRs and their mutants (MT) either expressed on primary T cells or hybridomas with CD3s, or purified ECD without CD3s, which form catch-slip bonds and slip-only bonds with their respective specific peptides presented by wild-type (WT) or MT MHCs, consisting of 42 datasets published by four labs and an additional dataset. We re-analyzed a TCR system published by the Zhu lab: the murine OT1 TCR expressed on either primary naive CD8$^+$ T cells, CD4$^+$CD8$^+$ thymocytes or soluble TCR ECD, which interacted with various peptides presented by a MT MHC (H2-K$^b\alpha$3A2) that abolished CD8 co-engagement[2,8] (nine datasets, Fig. 3a and Supplementary Fig. 1a). We also re-analyzed two TCR systems published by the Zhu lab and Chen lab: WT or two MT murine 2C TCRs either expressed on primary naive CD8$^+$ T cells, CD4$^+$CD8$^+$ thymocytes or CD8$^-$ hybridoma cells, which interacted with various peptides presented by H2-K$^b\alpha$3A2 (for CD8$^+$ primary T cells) or H2-K$^b$ (for CD8$^-$ hybridoma cells) without or with two point mutations specifically designed to alter bond profile, or by H2-L$^d$(m31), a different MHC allele from H2-K$^b$ (ref. 6) (19 datasets including soluble 2C TCR ECD, Fig. 3b and Supplementary Fig. 1b–d), and WT or three MT human 1G4 TCRs expressed on hybridoma cells, which interacted with the melanoma peptide NY-ESO-1 bound to HLA-A2 (ref. 6) (four datasets, Supplementary Fig. 1e). Furthermore, we re-analyzed five TCR systems published by the Evavold lab: the murine P14 TCR expressed on primary naive CD8$^+$ T cells, which interacted with various peptides presented by H2-D$^b$ with a D227K point mutation to abrogate CD8 binding[26] (three datasets, Fig. 3f and Supplementary Fig. 1f), and four mouse TCRs expressed on hybridomas interacted with NP$_{366}$ bound to the D227K mutant of H-2D$^b$ to prevent CD8 binding[13] (four datasets, Supplementary Fig. 1g). Moreover, we fitted our model to a TCR system published by the Lang lab: the soluble mouse N15 TCR ECD interacting with VSV and two MT peptides bound to H2-K$^b$ (ref. 4) (three datasets, Supplementary Fig. 1h). In addition, we performed an experiment specifically designed to test our model prediction that destabilizing the $\alpha_1\alpha_2$–$\beta_2$m interdomain bond of H2-K$^b$ would amplify TCR–pMHC catch bond (see Fig. 2a), which measured 2C TCR interaction with the same peptide (R4) presented by H2-K$^b\alpha$3A2 that had the WT mouse $\beta_2$m instead of the H2-K$^b\alpha$3A2 that swaps the mouse $\beta_2$m with the human $\beta_2$m (see below) (one dataset, Fig. 3b and Supplementary 1b). Gratifyingly, the theoretical reciprocal force-dependent off-rate $1/k(F)$ fits all 43 experimental bond lifetime vs force curves well (Fig. 3a, b, Supplementary Fig. 1, and Supplementary Tables 2 and 3), demonstrating our model's capability to describe a wide range of data.

**Characterization of force-lifetime relationship.** Previous work reported qualitative correlations between the TCR bond type, i.e., catch-slip bond vs slip-only bond, with the biological activity of the peptide to induce T-cell activation, i.e., pMHC potency[2,6–9]. To reduce data representation and extract more information quantitatively from the bond lifetime vs force data, we defined several metrics from their model fit for each TCR–pMHC system and examined their correlation with T-cell activation induced by a given interaction, using the OT1 system as an example because the quantitative ligand potency data are available[2,35]. We measured the peak bond lifetime, $t_{peak}$, and the change, $\triangle t$, from $t_{peak}$ to the force-free bond lifetime, $t_0 = 1/k_0$ (Fig. 3a, bond lifetime vs force profile of OT1 TCR:OVA:H2-K$^b$K$\alpha$3A2). We found the relative metric $\triangle t$ to be more suitable for comparison across different TCR systems, and to better correlate with ligand potency, than the absolute counterpart $t_{peak}$ (Fig. 3c). Although the force where catch-slip bond lifetime peaks, $F_{opt}$, occurs in a narrow range (10–20 pN), the force range, $F_{range}$, where bond lifetime returns from $t_{peak}$ back to $t_0$ defines the force span of a catch-slip bond over which force amplifies lifetime beyond $t_0$ (Fig. 3a, bond lifetime vs force profile of OT1 TCR:OVA:H2-K$^b$K$\alpha$3A2). Both scaled parameters, $L = \triangle t / t_{peak}$ (relative length of lifetime) (Fig. 3d, relative length ($L$)), and to a lesser extent $B = (F_{range} - F_{opt}) / F_{opt}$ (relative breadth of lifetime) (Supplementary Fig. 2a), correlate with ligand potency well. We define a scaled parameter, $I = L/(1 + B)$, which is the area ratio of two rectangles: $\triangle t \times F_{opt}$ over $t_{peak} \times F_{range}$. Remarkably, this combined parameter, which we term the catch-bond intensity or catchiness, correlates best with the ligand potency across different TCR systems (Intensity of catch bond ($I$) in Fig. 3d, e), supporting its usefulness as a metric of reduced data representation for a bond profile.

**Model parameters' correlation to ligand potency.** It seems reasonable to test the validity of our model by examining the possible correlation of (or the lack thereof) the model parameters with features of the biological system, e.g., the ligand potency. The rationale is that if its parameters are capable of capturing and predicting such biological features, then the model would be more meaningful and useful than merely a curve-fitting tool. Therefore, we plotted the tilted angle of the bonding interface $\theta$, the number of the unfolded amino acids $n^*$, and the width of the zero-force free-energy well $\delta_0^*$ that best-fit the force-lifetime curves of OT1, 2C, P14, N15, and TRBV TCRs interacting with their corresponding panels of pMHCs (Fig. 3e). Gratifyingly, we observed good correlation between each model parameter and the peptide potency for all 21 published datasets of TCR–pMHC-I catch-slip bonds and slip-only bonds measured by four independent laboratories in five papers[2,4,6,13,26]. Moreover, model parameter and the peptide potency for OT1 quantitatively showed positive correlation with linear fitting (Fig. 3f).

In a previous study, we mutated residues in the 2C or 1G4 TCR and/or their corresponding pMHCs to alter bond profiles as predicted by SMD simulations, which was confirmed by BFP experiment[6]. We therefore fitted our model to the force-lifetime curves of these mutant TCR–pMHC interactions to evaluate the model parameters, $\delta_0^*$, $\theta$, and $n^*$ (Fig. 4a and Supplementary Fig. 2b, c). In the absence of other functional data, we took an indirect approach to examine their correlations with the catchiness $I$ of these bond lifetime vs force curves (Fig. 4b) since $I$ and all three model parameters correlate with the peptide potency (Fig. 3e, f). Results are exemplified by the $\delta_0^*$, $\theta$, and $n^*$ vs $I$ plots, which are graphed together with the data without TCR and MHC mutations that already showed functional correlates. For the WT OT1, 2C, P14, N15, and TRBV TCRs interacting with their corresponding panels of pMHCs, the best-fit model parameters $\delta_0^*$ (Fig. 4c), $\theta$ (Supplementary Fig. 2d), and $n^*$ (Supplementary Fig. 2e), correlate with the peptide potency predictor $I$ (blue-open symbols). Remarkably, for the 2C and 1G4 TCRs specifically mutated to alter bond profiles with the corresponding WT or MT MHCs presenting the same agonist

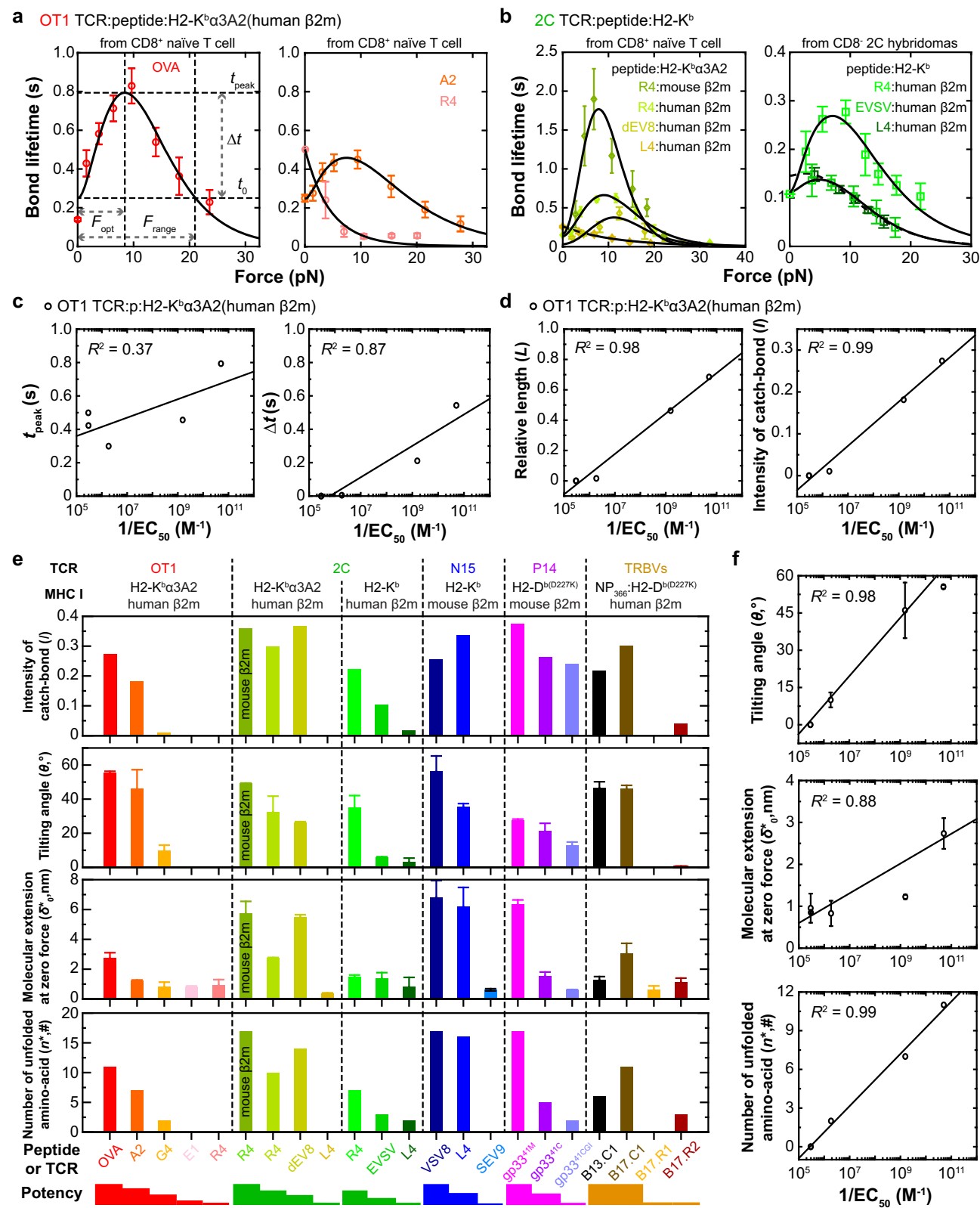

peptide, their best-fit $\delta_0^*$, and to a lesser extent, $\theta$ and $n^*$, also correlate well with $I$ (Fig. 4c and Supplementary Fig. 2d, e, green-closed symbols). Interestingly, $1/k_0$ shows no correlation with $I$ (Supplementary Fig. 2f), consistent with reports that zero-force bond lifetime does not correspond to ligand potency in these cases[2,6,35].

It is worth pointing out that the above results not only support our model's validity but they also suggest that our model is more than a mere analytical framework to organize experimental data. Rather, the model parameters may be used to distinguish antigen recognition efficacy with force-amplified discriminative power. For example, the correlations of $\theta$ and $\delta_0^*$ with peptide potency (Figs. 3f and 4c) indicate that the more potent the peptide, the higher the force sensitivity of its TCR–pMHC interaction, and the narrower the force range over which the TCR–pMHC interaction is sensitive to force (Fig. 2g).

**Fig. 3 | TCR bond type characterization and correlation with pMHC-I biological activity. a, b** Fitting of theoretical $1/k(F)$ curves to experimental bond lifetime vs force data (points, mean ± SEM from $n > 50$ bond lifetime data per each force bin, re-analyzed from refs. 2,6) of OT1 (**a**) or 2 C (**b**) TCR expressed on CD8+ naive T cells interacting with indicated p:H2-Kb α3A2 ((**a**) and (**b**) left) or on CD8− 2C hybridomas interacting with indicated p:H2-Kb ((**b**) right). Several metrics are defined to characterize the force-lifetime curve as indicated in the left panel of (**a**): $F_{opt}$ is the "optimal force" where lifetime peaks ($t_{peak}$), $\triangle t$ is the lifetime increase from the zero-force value $t_0$ to $t_{peak}$, and $F_{range}$ is the range over which force amplifies lifetime beyond $t_0$. **c, d** Two-dimensional metrics, $t_{peak}$ and $t$ (**c**), and two dimensionless metrics, $L = \triangle t/t_{peak}$ and $I = L/(1 + B)$ where $B = (F_{range} - F_{opt})/F_{opt}$ (**d**), are plotted vs the logarithm of the reciprocal peptide concentration required to stimulate half-maximal T-cell proliferation ($1/EC_{50}$) and fitted by a linear function. **e** A single-valued catch-bond intensity $I$ (1st row), best-fit model parameters $\theta$ (tilted angle of the bonding interface, 2nd row), $\delta_0^*$ (width of zero-force free-energy well, 3rd row), and $n^*$ (number of unfolded amino acids, 4th row) derived from the fitted force-lifetime curves of OT1, 2C TCR on primary T cells, 2C TCR on hybridomas, purified N15 TCRαβ, P14 TCR on primary T cells, or TRBV TCRs (B13.C1/B17.C1 and B17.R1/B17.R2) expressed on hybridomas interacting with their corresponding pMHCs are plotted according to the ranked-order of peptide potencies (bottom). All error bars present standard error (SE) derived from fitting of the model to mean ± SEM of bond lifetimes (Supplementary Table 3). **f** Best-fit model parameters $\theta$ (the tilted angle of the bonding interface, 1st row), $\delta_0^*$ (the width of zero-force free-energy well, 2nd row), and $n^*$ (the number of unfolded amino acids, 3rd row) are plotted vs the logarithm of the reciprocal peptide concentration required to stimulate half-maximal T-cell proliferation ($1/EC_{50}$) and fitted by a linear function. All error bars represent SE derived from fitting of the model to mean ± SEM of bond lifetimes (Supplementary Table 3). Source data are provided as a Source Data file.

**Comparison between coarse-grained and all-atom models.** Bonding interface tilting has been observed to be associated with changes in the number of hydrogen bonds bridging the TCR and pMHC molecules as they were pulled to unbind in SMD simulations[6]. Therefore, we investigated whether, and if so, how well the tilting angle would correlate with the change of hydrogen bonds between TCR and pMHC. Remarkably, $\theta$ was found to be proportional to the net change in the total number of hydrogen bonds at the bonding interface (Fig. 4d and Supplementary Fig. 3). This finding is intuitive and supports the validity of our coarse-grained model because it is able to recapitulate the results of all-atom SMD simulations[6].

**Classification of bond types by clustering analysis on phase diagrams.** In Fig. 2e–g, we have explored the model parameter space to identify regions that correspond to slip-only bonds and catch-slip bonds. Here we examined whether, and if so, how parameters that best-fit different experimental bond types map onto different regions of the parameter space. Since the model has four parameters, $\theta$, $\delta_0^*$, $d_{\alpha3}$, and $n^*$ ($k_0$ is not considered because if its lack of correlation with catch-bond intensity), we analyzed their clustering and projected their values in the 4D parameter space onto three phase diagrams spanning the $\theta$-$\delta_0^*$ (Fig. 4e), $\theta$-$d_{\alpha3}$ (Fig. 4f and Supplementary Fig. 4), and $\delta_0^*$-$n^*$ (Fig. 4g) 2D space. Clustering analysis of the model parameters that best-fit 43 TCR–pMHC bond lifetime vs force curves (Supplementary Fig. 5) shows three distinct clusters in the $\delta_0^*$ vs $\theta$ and $\theta$ vs $d_{\alpha3}$ plots as well as $\delta_0^*$ and $\theta$ vs $n^*$ plots (Fig. 4e–g), which classify the TCR–pMHC interactions into slip-only (SO), weak catch-slip (WC) and strong catch-slip (SC) bonds, which correspond to weak, intermediate, and strong potencies for pathogenic peptides and their variants. Whereas transition in bond type from SO to WC and SC requires monotonical increase in $\theta$ and $n^*$ (Fig. 4f, g), the corresponding change in $\delta_0^*$ is non-monotonic (Fig. 4e–g). SO bonds show small $n^*$, $\delta_0^*$, and $\theta$ values. WC and SC bonds observed from experiments are best-fitted by similar $n^*$ (9 for WC and 11 for SC) but oppositely ranked $\delta_0^*$ and $\theta$ values. To change from WC to SC bonds requires a slight increase in $\delta_0^*$ (from 3 to 3.5 nm) and a large increase in $\theta$ (from 20 to 45°) (Fig. 4g). We also performed principal component analysis and calculated the Mahalanobis distances of the principal axes for the three bond types[36], which are statistically separated in the catch-bond intensity vs Mahalanobis distance plot (Fig. 4h). Interestingly, WC and SC bonds show distinct conformational changes despite their similar $I$ values measured from the force-lifetime curves. The corresponding structural features of these three types of bonds are depicted in Fig. 4i, which have been observed in our previous SMD studies[6]. Of note, model parameters visualized by SMD simulations are usually larger than their best-fit values, which may have two explanations: First, to enable dissociation to be observed in affordable computational times, much higher forces were used in simulations than experiments to accelerate the biophysical processes, which likely induced much larger conformational changes. Second, our model describes the average conformational change during the entire dissociation process, which is smaller than the maximum conformational changes likely to occur right before unbinding and to be captured by SMD.

**Our structure-based model is superior to the generic two-pathway model.** It seems that other published catch-bond models should also be able to fit the experimental force-lifetime profiles analyzed here, given their relatively simple shapes. As an example, we examined the two-pathway model below[24]:

$$k(F) = k_{0,p1}e^{\delta_{0,p1}^* F/k_B T} + k_{0,p2}e^{\delta_{0,p2}^* F/k_B T} \tag{4}$$

where $k_{0,p1}$ and $k_{0,p2}$ are the respective zero-force off-rates of the first and second pathway, $\delta_{0,p1}^*$ and $\delta_{0,p2}^*$ are the respective distances from the bound state to the transition states along the first and second pathways (Fig. 1a). Here, the off-rate for each pathway takes the form of the Bell model, but the catch pathway parameter $\delta_{0,p1}^*$ has a negative value[24] (Supplementary Table 4). This model is generic as it has previously been applied to TCR–pMHC catch-slip bonds without considering the specific conformational changes[4].

As expected, Eq. (4) also fitted our experimental data with goodness-of-fit measures statistically indistinguishable to Eq. (2) (Supplementary Figs. 1 and 10a, b). However, the fitting parameters correlate with neither the TCR/pMHC-I potency to induce T-cell function nor the catch-bond intensity (Supplementary Fig. 6, see the negative or zero correlation and the poor $R^2$ values), hence have no biological relevance. This comparison indicates that the model developed herein is superior to the previous two-pathway model.

**Model for TCR catch bonds with class II pMHC**

MHC class II differs from class I in three main aspects (comparing Fig. 2a and Fig. 5a): (1) MHC-I has three α domains and a $\beta_{2m}$ domain whereas MHC-II has two α and two β domains. (2) MHC-I anchors to the T-cell membrane through a single linker to the $\alpha_3$ domain. The $\beta_{2m}$ domain attaches to the $\alpha_3$ domain instead of anchoring to the T-cell membrane directly. By comparison, MHC-II anchors to the membrane through two linkers, one to the $\alpha_2$ domain and the other to the $\beta_2$ domain. (3) The peptide is presented by the $\alpha_1$-$\alpha_2$ domains of MHC-I but the $\alpha_1$-$\beta_1$ domains of MHC-II. These structural differences alter how forces are supported by and transmitted through, and induce conformational changes in, the TCR complexes with pMHC-I vs pMHC-II. Thus, it is necessary to modify the previous model in order for it to describe TCR catch and slip bonds with pMHC-II, which is done by using a different $\delta_1\gamma(F)$ expression than Eq. (3) (Supplementary Model Derivations, Section B). This modification assumes force-induced partial unfolding and stretching of the TCR Vα-Cα joint and the MHC $\alpha_1$-$\alpha_2$ and $\beta_1$-$\beta_2$ joints during dissociation, which results in tilting of the bonding interface (Fig. 5a, b).

In the class II model, the same parameters $\delta_0^*$, $n^*$, and $\theta$ are used but the MHC contribution to $n^*$, i.e., $n_{p,MHC}$, represents the average

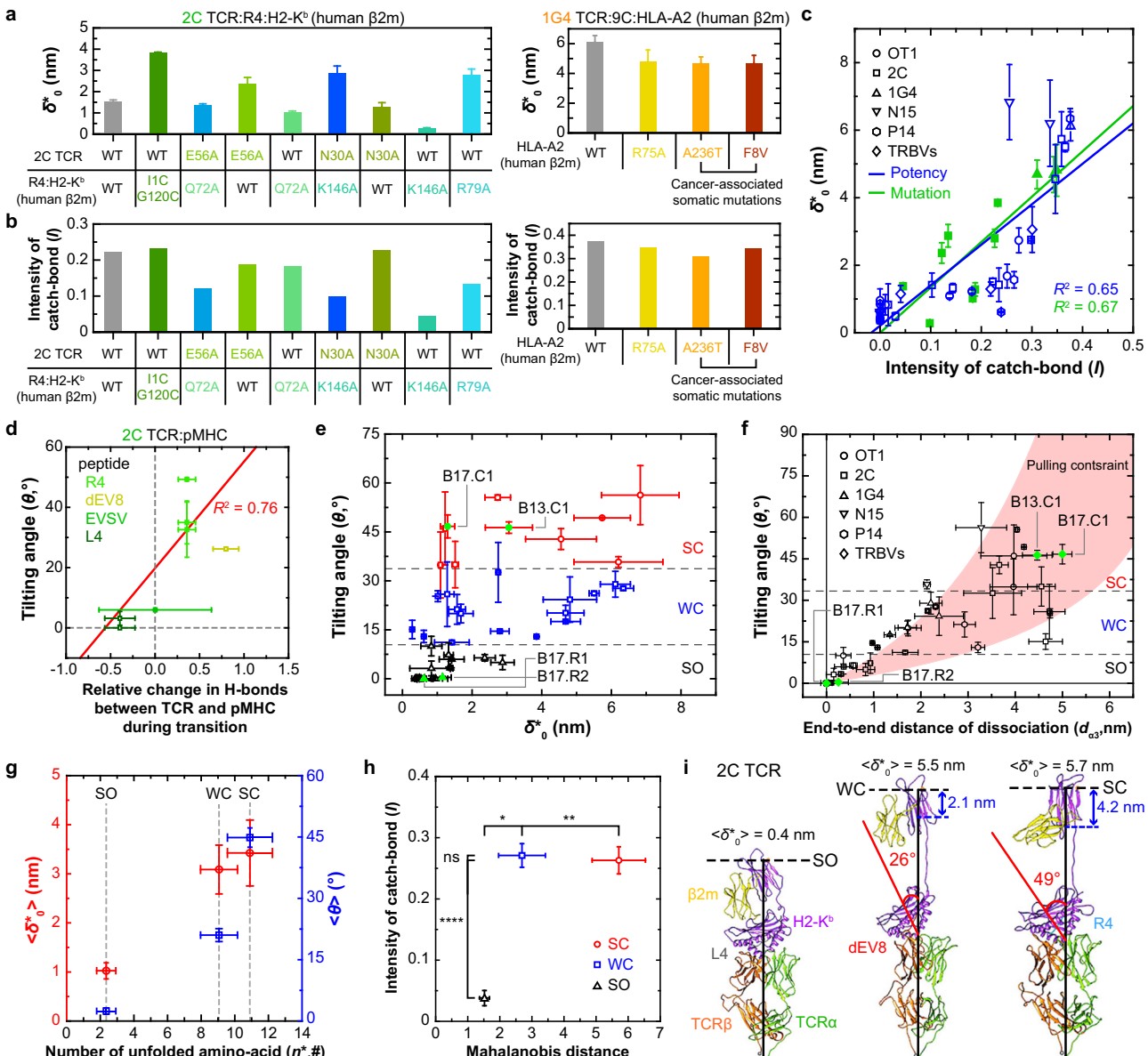

**Fig. 4 | Properties and biological relevance of class I model parameters. a, b** The width of zero-force energy well $\delta_0^*$ (**a**) and the single-valued catch-bond intensity $I$ (**b**) calculated from WT or mutant 2C TCRs (left) and WT 1G4 TCR (right) interacting with their corresponding WT or MT pMHCs. The MT 2C TCRs and H2-K$^b$s were designed to destabilize the TCR–pMHC interaction. The MT p:HLA-A2s were designed to either destabilize the TCR–pMHC interaction (R75A) or stabilize the MHC intramolecular interaction (A236T and F8V). All error bars represent SE derived from fitting the model to mean ± SEM of bond lifetimes. **c** Data (presented as the best-fitting value ± SE) from Fig. 3e (3rd row) are re-graphed as $\delta_0^*$ vs $I$ plot to show their correlation (blue). Additional $\delta_0^*$ vs $I$ data from MT TCRs and/or MT pMHCs without functional data also show strong correlation (green). Different TCR systems are indicated by different symbols. The two datasets were separately fitted by two straight lines with the goodness-of-fit indicated by $R^2$. **d** Tilting angle of the bonding interface ($\theta$) vs normalized net gain of hydrogen bonds at the interface between 2C TCR and the indicated pMHCs is plotted (points) and fitted (line) (error bars in $x$- and $y$ axes represent SD from Supplementary Fig. 3 and SE of $\theta$, respectively). **e** Clustering analysis shows three clusters in the $\delta_0^*$-$\theta$ phase diagram: slip-only (SO, black), weak catch-slip (WC, blue), and strong catch-slip (SC, red) bonds. Data indicate the best-fitting value ± SE. **f** Tilting angle ($\theta$) vs end-to-end distance of dissociated α3 domain ($d_{\alpha3}$). The three types of bonds, SO, WC, and SC, are also

clustered in this phase diagram, which are separated by the dotted lines that predicted from the pulling constraints of the model. The two pairs of TRBV TCRs are indicated in e and f by green dots. Data indicate the best-fitting value ± SE. **g** The average molecular extensions at zero force ($\langle \delta_0^* \rangle$, left ordinate) and the average rotation angle ($\langle \theta \rangle$, right ordinate) (mean ± SEM) are plotted vs the total number of unfolded amino acids ($n^*$, abscissa) to show three clusters. Each bond type is indicated by a dotted line ($n = 10$, 16, and 17 for numbers of data in the SO, WC, and SC groups, respectively; individual data of each cluster are shown in (**e**, **f**)). **h** Catch-bond intensity vs Mahalanobis distance plot (mean ± SEM), again showing three clusters. Principal component analysis was used to find principal axes. Mahalanobis distances for each cluster were calculated using common principal axes from total dataset (numbers of data are the same as (**g**)). ****$P < 0.0001$; **$P < 0.01$; *$P < 0.05$, and ns $> 0.05$ by one-sided unpaired $t$ test. **i** Structural models illustrating the conformations of three bond types according to their model parameters based on the previous SMD simulation of the 2C TCR system[6]. Two structural parameters ($\theta$, red; $d_{\alpha3}$, blue) are indicated to show the differences between bond types. Unless otherwise described, all errors shown in (**a**–**f**) are SE derived from fitting the model to mean ± SEM of bond lifetimes (Supplementary Table 3). Source data are provided as a Source Data file.

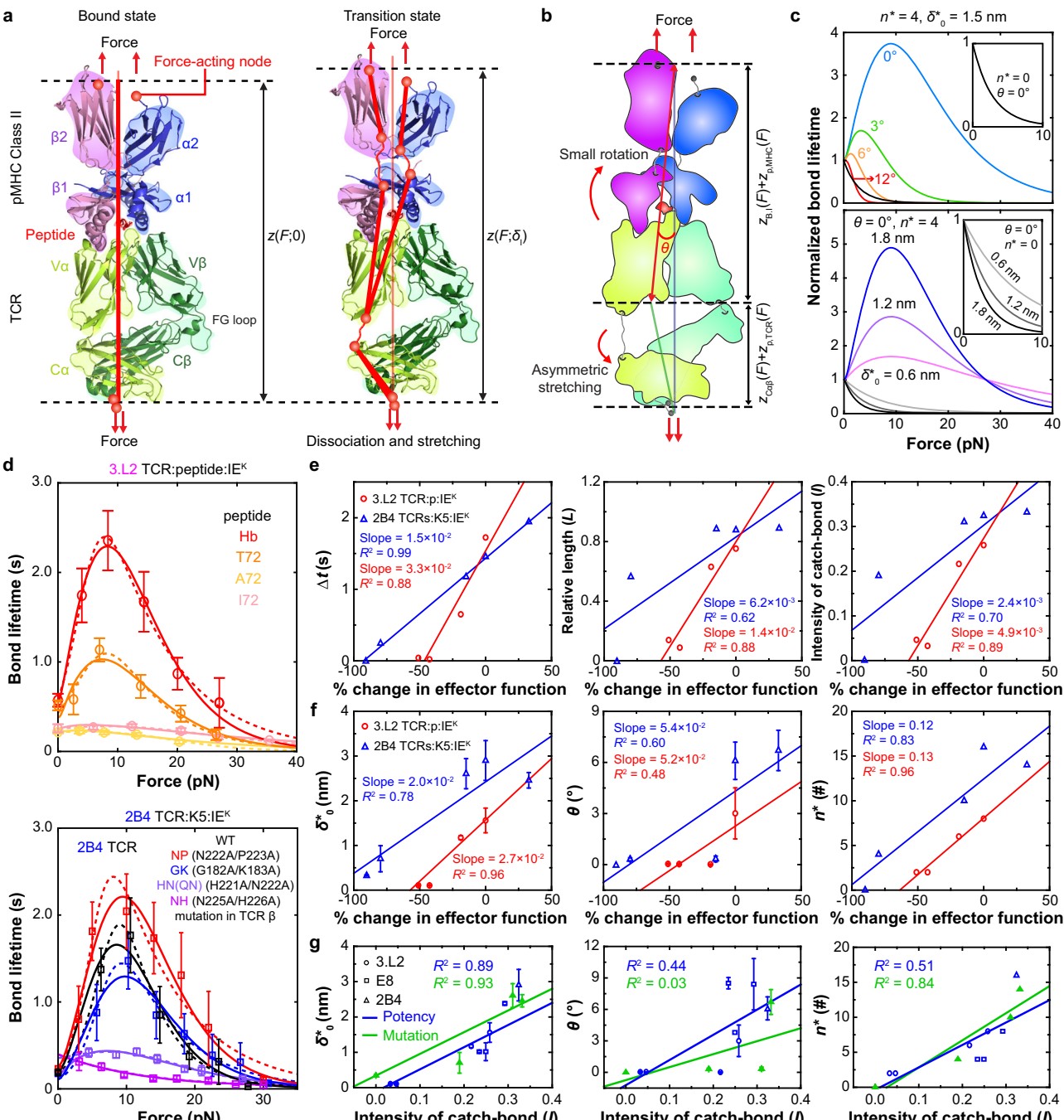

**Fig. 5 | The TCR−pMHC-II model. a** Force-induced conformational changes of a TCR−pMHC-II complex as it traverses from the bound state (*left*) to the transition state (*right*). The diagrams are based on the published co-crystal structure (2IAM) of the E8 TCR α (yellow) β (green) subunits and the TPI peptide (red) bound to the HLA-DR1 α (blue) β (pink) subunits with various domains indicated. The force-transmission paths are shown as red lines connecting the force-acting nodes. **b** Various contributions to the total extension projected on the force axis: stretching of the TCR Cα and Cβ domains ($z_{C\alpha\beta}$), asymmetric partial unfolding of the TCR Vα−Cα and Vβ−Cβ interdomain hinges ($z_{p,TCR}$), asymmetric partial unfolding of the MHC α1−α2 and β1−β2 interdomain hinges ($z_{p,MHC}$), and rotation between the α1−β1 and α2−β2 domain hinges and tilting of the bonding interface between the MHC α1−β1 and the TCR Vα−Vβ by an angle θ ($z_{B,I}$). **c** Theoretical normalized bond lifetime vs force curves. The effects of changing θ and $\delta_0^*$ are shown in the upper and lower panels, respectively, for the indicated parameter values. **d** Fitting of predicted $1/k(F)$ curves (dashed lines by two-pathway model and solid lines by TCR−pMHC-II model) to experimental bond lifetime vs force data

(points, mean ± SEM from >50 bond lifetime measurements per force bin) of 3.L2 TCR on CD4⁻CD8⁺ naive T cells interacting with indicated p:I-Eᵏ's[3] (upper) or WT and indicated mutant 2B4 TCRs on hybridomas interacting with K5:I-Eᵏ. **e** Dimensional metrics, *t* (left), scaled relative length of bond lifetime *L* (middle), and intensity of catch bond *I* (right) vs reciprocal % change (relative to WT) of effector function, i.e., the peptide dose required for 3.L2 T cells to generate 40% B cell apoptosis (1/EC₄₀)[37] (red) or the area under the dose response curve (AUC) of the 2B4 hybridoma IL-2 production[38] (blue) plots. **f** Best-fit model parameters $\delta_0^*$ (left), θ (middle), and $n^*$ (right) are plotted vs reciprocal relative % change of effector function. **g** The three model parameters in f for both the 3.L2 and 2B4 TCR systems are plotted vs the catch-bond intensity *I* and fitted by a straight line. We also added to each panel an additional point obtained from data and model fit of E8 TCR−TPI:HLA-DR1 interactions[10]. All errors in (**f**, **g**) are SE derived from fitting the model to mean ± SEM of bond lifetime (Supplementary Table 6). Source data are provided as a Source Data file.

number of amino acids in the polypeptides of the partially unfolded MHC-II $\alpha_1$–$\alpha_2$ and $\beta_1$–$\beta_2$ joints instead of the MHC-I $\alpha_1\alpha_2$–$\alpha_3$ joint, and the relationships between $\theta$ to other structural parameters are also different from the class I model (Fig. 5b and Supplementary Model Derivation, Section B). Like the class I model, the $k_0/k$ vs $F$ plots for a range of $n^*$, $\theta$, and $\delta_0^*$ in Fig. 5c show similar features to Fig. 2e and meet our objective of being capable of describing catch-slip bonds if and only if $n^* > 0$ and $\theta \geq 0$. Unlike the class I model, a much smaller $\theta$ value (<10°) is seen in the class II model (compared Fig. 5c and Supplementary Fig. 7 with Fig. 2e–g), indicating the main conformational change responsible for TCR–pMHC-II catch-slip bond is unfolding rather than tilting. The validity of this model is supported by its excellent fitting to our 6 published datasets of mouse 3.L2 (Fig. 5d, 3.L2 TCR:peptides:I-E$^k$)[3] and human E8 (Supplementary Fig. 9i)[10] TCRs.

In addition, we generated five additional datasets in this work specifically designed to test our model prediction that (de)stabilizing the TCR–CD3 complex would alter the TCR–pMHC bond profile (Supplementary Table 6). Of these, the WT represents a hybrid 2B4 TCR with its mouse V$\alpha\beta$ fused with the C$\alpha\beta$ of the human LC13, expressed on hybridoma cells with human CD3 (see below) and the four double mutants each replaces two C$\beta$ residues by Ala to respectively decrease (NP) or increase (GK, HN, and NH) C$\beta$–CD3 interactions under force (see below). Remarkably, interactions of the same K5:I-E$^k$ with these five TCRs indeed yielded different bond profiles that were well fitted by our class II model (Fig. 5d, 2B4 TCRs:K5:I-E$^k$).

Furthermore, the four metrics $\triangle t$, $L$, $I$, and $t_{\text{peak}}$ of both the 3.L2 and 2B4 TCR–pMHC-II bond lifetime vs force curves correlate well with the published peptide (for 3.L2) and TCR (for 2B4) potencies[37,38] (Fig. 5e and Supplementary Fig. 8). Moreover, the three model parameters $\theta$, $n^*$, and $\delta_0^*$ also correlate well with the TCR potency for the 2B4 system[38] and with the ligand potency for the 3.L2 system[3,37] (Fig. 5f, see the goodness of fitting, $R^2$), supporting the ability of the metrics of the bond profile and the model parameters to recognize the change in the TCR–CD3 ECD interaction in addition to the ability to discriminate antigen. These properties are desirable, intuitive, and are consistent with the parallel properties found in the class I model. Similar to the class I model parameters, $\delta_0^*$ correlates well with the catch-bond intensity for the pooled results from all class II data (Fig. 5g, $\delta_0^*$ vs $I$), but $\theta$ and $n^*$ correlate less well with $I$ (Fig. 5g, $\theta$ vs $I$ and $n^*$ vs $I$). Thus, the validity of the class II model is further supported by the faithful mapping of the relationship between biophysical measurements of catch and slip bonds and biological activities of the TCR–pMHC-II interactions onto a relationship between model parameters and biological function.

As expected, Eq. (4) also fitted our TCR–pMHC-II data with goodness-of-fit measures statistically indistinguishable to Eq. (2) (Fig. 5d and Supplementary Fig. 10c, d). Similar to the class I system, the fitting parameters of Eq. (4) correlate with neither the TCR/pMHC-II potency to induce T-cell function nor the catch-bond intensity (Supplementary Fig. 6, see the negative or zero correlation and the poor $R^2$ values), hence have no biological relevance. This comparison again indicates that the model developed herein is superior to the previous two-pathway model.

## Cross-examination of class I model against class II data and vice versa

Upon examining the catch-slip and slip-only bond lifetime vs force curves in Figs. 3a, b and 5d and Supplementary Fig. 1, it became apparent that the data seem very similar regardless of whether they are for class I or class II pMHC. Indeed, applying the class I model to the class II data and vice versa indicates that both models are capable of fitting both data well (Supplementary Fig. 9) and produce statistically indistinguishable goodness-of-fit measures (Supplementary Fig. 10). This is not surprising because both models have five fitting parameters and the bond lifetime vs force curves have relatively simple shapes.

Nevertheless, fitting the same data by different models returns different parameter values depending on the model used, because the two models are constructed based on different structures and force-induced conformational changes of the TCR–pMHC complexes. Therefore, we asked whether the best-fit model parameters were capable of distinguishing data from the two classes of pMHCs and of telling whether a correct model was used to analyze data of matched MHC class. To answer these questions, we plotted $\delta_0^*$ vs $I$ (Fig. 6a, b) and $\delta_0^*$ vs $n^*$ (Fig. 6d, e) using values of the two models that best-fit the data of OT1, 2 C, 1G4, P14, N15, and TRBV TCRs interacting with their respective panels pMHC-I ligands (Fig. 6a, d) as well as 3.L2, WT and MT 2B4, and E8 TCRs interacting with their respective panels of pMHC-II ligands (Fig. 6b, e). Surprisingly, the dependency of $\delta_0^*$ on $I$ is 2–5-fold stronger (i.e., steeper slope) (Fig. 6c), indicating a greater discriminative power of receptor/ligand potency, for the matched than the mismatched cases. Furthermore, it is well-known that the average contour length per a single amino acid $l_c$ is ~0.4 nm[19,39,40], which sets the biophysical limit for the slope of $\delta_0^*$ vs $n^*$ plots. Indeed, we found that the slopes of the $\delta_0^*$ vs $n^*$ plots are within this limit for both model fits of both class I and class II data (Fig. 6f). Moreover, the goodness-of-fit ($R^2$) values of the linear fit to the $\delta_0^*$ vs $I$ (Fig. 6c) and $\delta_0^*$ vs $n^*$ (Fig. 6f) data are much greater for the matched than the mismatched cases, indicating more appropriate models for the data in the matched than the mismatched cases. Indeed, the $R^2$ value for fitting the class II data by the class I model is too small to be statistically reasonable, therefore telling the mismatch between the model and the data. These results indicate that the model parameters are capable of distinguishing data from the two classes of pMHCs.

## Model validation by mutagenesis to alter force-induced conformational changes

The published datasets re-analyzed by our models include TCR interactions with altered peptide ligands, yielding different catch and slip bonds whose profile metrices and model parameters correlate with varied peptide potencies to induce T-cell activation (Figs. 3, 3.L2 TCR:peptides:I-E$^k$ of Fig. 5d–f red, and 5g blue; Supplementary Figs. 1a, 2C TCR:peptides:H2-K$^b$ (or H2-K$^b\alpha$3A2) of Fig. 1b–d, 1G4 TCR:9C:HLA-A2 of 1e–h, 8 red, and 9i). They also include mutations on the TCR or MHC specifically designed to assess how structural change altered bond profile (mutant 2C TCRs:R4:mutant H2-K$^b$s in Supplementary Fig. 1b, and 1G4 TCR:9C:mutant HLA-A2s in 1e) but do not include functional data[6]. We thus performed two sets of new studies to further validate the class I and II models, respectively, using mutations located away from the TCR and pMHC binding interface but capable of impacting their respective conformational changes under force, which were analyzed by MD simulations, bond lifetime measurements, and functional assays.

The first set of studies compared the H2-K$^b$ with a WT mouse $\beta_2$m and a H2-K$^b$ that swaps the mouse $\beta_2$m with the human $\beta_2$m because the latter binds the mouse class I heavy chain with a higher affinity and better support peptide binding than the former[41]. Since it is easier to make soluble H2-K$^b\alpha$3A2 protein with a human than mouse $\beta_2$m, many of our previous bond profile measurements used the former protein (Supplementary Table 3). Surprisingly, T cells kill less efficiently target cells expressing the H2-K$^b$ with a human than mouse $\beta_2$m[42]. Our previous study using double-cysteine mutations to lock the $\alpha_1\alpha_2$–$\beta_2$m connection by disulfate bond suppressed both pMHC conformational changes and its catch bond with TCR concurrently[6] (Supplementary Fig. 1b, compared the R4 curves in panels 2 and 3). Using SMD simulations, we observed force-induced dissociation of the $\alpha_1\alpha_2$–$\beta_2$m interdomain bond (Supplementary Movie 1). We compared MD simulated interactions of H2-K$^b$ $\alpha$ chain with mouse $\beta_2$m (using the crystal structure 1G6R) and human $\beta_2$m (using a model built based on 1G6R and 2BNR), finding that Arg14, Glu232, and Gly237 of the H2-K$^b$ $\alpha$ chain respectively interacted with three residues—Asp34, Lys6, and

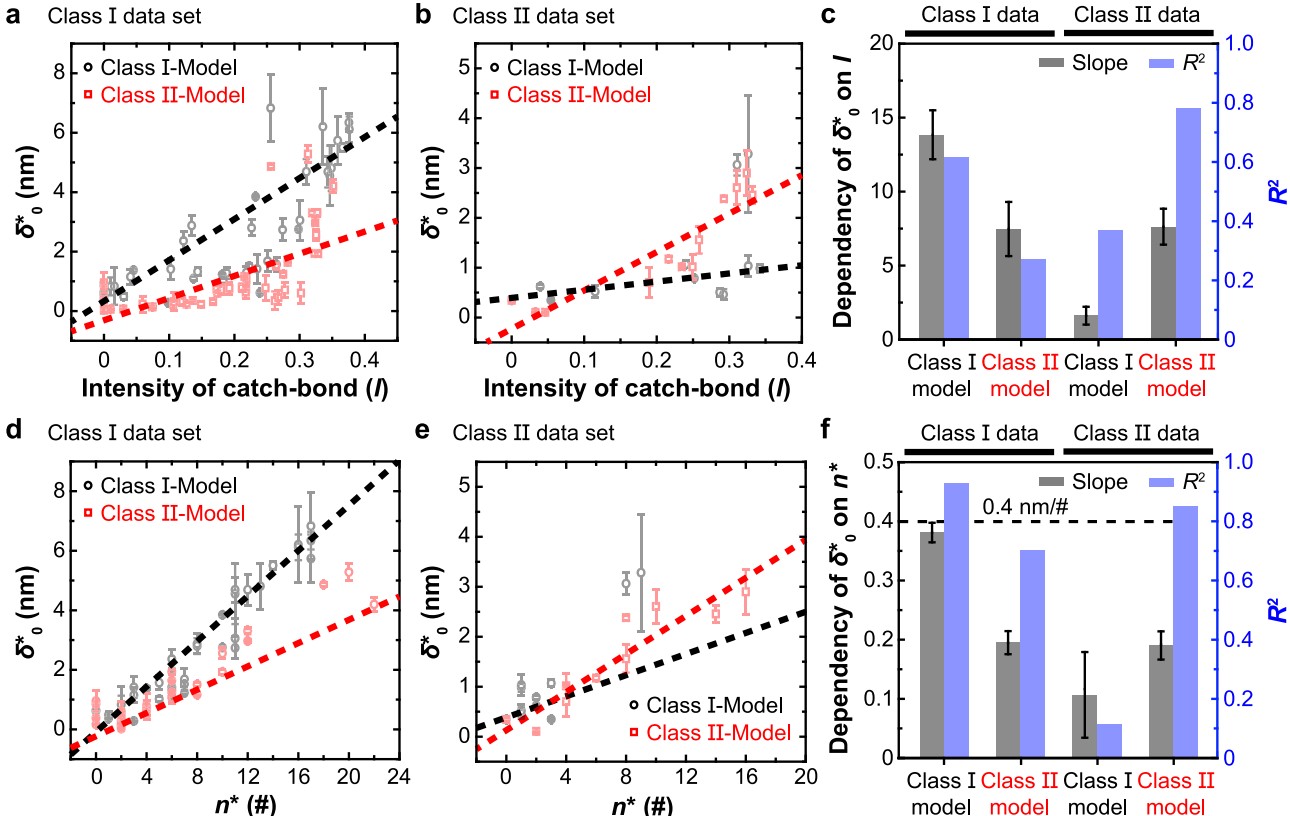

**Fig. 6 | Cross-examination of class I and II models against class I and II data.** **a**, **b** $\delta_0^*$ vs $I$ plots (data presented with the best-fitting parameters with SE) obtained using class I (black) or class II (red) model to fit force-lifetime data of TCR interacting with pMHC-I (**a**) or pMHC-II (**b**) molecules. In each panel, two sets of parameter values were returned from fitting depending on whether class I (black) or class II (red) model was used because they are based on different structures and conformational changes of the TCR–pMHC complexes. **c** The slopes (gray, the level of correlation between $\delta_0^*$ and $I$) and goodness-of-fit ($R^2$) (blue, the degree of appropriateness of the model for the data) of the linear fit in (**a**, **b**) are shown in the matched (1st and 4th groups) and mismatched (2nd and 3rd groups) cases. All error

bars represent SE of linear fitting. **d**, **e** $\delta_0^*$ vs $n^*$ plots (data presented with the best-fitting parameters with SE) obtained using class I (black) or class II (red) model to fit force-lifetime data of TCR interacting with pMHC-I (**d**) or pMHC-II (**e**) molecules. **f** The slopes and goodness-of-fit of the linear fit in (**d**, **e**) are shown in the matched (1st and 4th groups) and mismatched (2nd and 3rd groups) cases. The slopes indicate the average unfolding extension per amino acid (nm/a.a.) from each model, which are compared to the maximum average contour length per amino acid of ~0.4 nm/a.a. (biophysical limit, black dashed line with considerable deviation)[19,39,40]. All error bars represent SE of linear fitting. Source data are provided as a Source Data file.

Tyr67—of the human $\beta_2$m but not the corresponding residues of the mouse $\beta_2$m (Fig. 7a–c). This indicates that the hybrid H2-K$^b$ has a more stable structure and hence less able to respond to force induction of conformational change than the WT molecule, predicting a less pronounced TCR catch bond with the same peptide presented by the hybrid than the WT H2-K$^b$. Remarkably, the newly measured force-dependent bond lifetime indeed showed a much more pronounced catch bond of the 2C TCR with R4 peptide bound to H2-K$^b\alpha$3A2 with a mouse $\beta_2$m than hybrid H2-K$^b\alpha$3A2 with a human $\beta_2$m (Fig. 3b, 1st panel), supporting the prediction of our class I model. Consistent with previous report[42], functional assay also showed that the WT H2-K$^b$ with a mouse $\beta_2$m was more able to activate T cells than hybrid H2-K$^b$ with a human $\beta_2$m (Fig. 7d, e and Supplementary Fig. 11), further validating the class I model.

The second set of studies examined a hybrid TCR with the mouse 2B4 V$\alpha\beta$ fused with the human LC13 C$\alpha\beta$ and 4 double mutations on the C$\beta$ domain, which have been indicated by our previous NMR and chemical shift experiments[38] and by recently published cryoEM structures[43,44] to impact its interactions with human CD3 (Fig. 7f). We performed MD simulations to examine the C$\beta$–CD3 *cis*-interactions in the absence (Fig. 7g) and presence (Fig. 7h) of force to mimic pulling on the V$\alpha\beta$ by the engaged K5:I-E$^k$ (Supplementary Movies 2–4). We found that C$\beta$ Pro223 is force-stabilizing (Fig. 7i, distance between P223 in TCR$\beta$ and L90 in CD3$\epsilon$) whereas C$\beta$ Lys183 and Asn225 are force-destabilizing (Fig. 7i, distance between K183 (in TCR$\beta$) and L90

(in CD3$\epsilon$‘), and distance between N225 (in TCR$\beta$) and E38 (in CD3$\gamma$), respectively). These results suggest that the double mutation N222A/P223A (NP) may result in less stable, whereas G182A/K183A (GK) and N225A/H226A (NH) may result in more stable, C$\beta$–CD3 *cis*-interactions under force, therefore potentially limiting force-induced conformational changes in the TCR$\alpha\beta$ less (for NP) and more (for GK and NH), respectively, than the WT molecule. Interestingly, NP was identified as a gain-of-function mutation whereas GK and NH (plus another double mutant H221A/N222A, or HN) were identified as loss-of-function mutations by functional assays[38]. Supporting the prediction of our class II model, force-dependent bond lifetime measurements by BFP indeed showed a more pronounced catch-slip bond of the NP mutant, and less pronounced catch-slip bonds of the GK and HN mutants, than the WT 2B4 TCR interaction with K5:I-E$^k$ (Fig. 5d, 2B4 TCRs:K5:I-E$^k$). Another mutant, NH, showed reduced function in IL-2 production and BFP experiment found slip bond. Remarkably, the bond profile metrices (Fig. 5e and Supplementary Fig. 8, blue lines) and best-fit model parameters (Fig. 5f, blue line and 5g, green line) were found to correlate with T-cell function, further validating the class II model.

## Discussion

With the exception of a recent paper that observed T-cell pulling on TCR by ~2 pN forces using a spider silk peptide-based force probe[45], five publications from two laboratories demonstrated that TCR experienced endogenous forces of 12–19 pN using DNA-based force

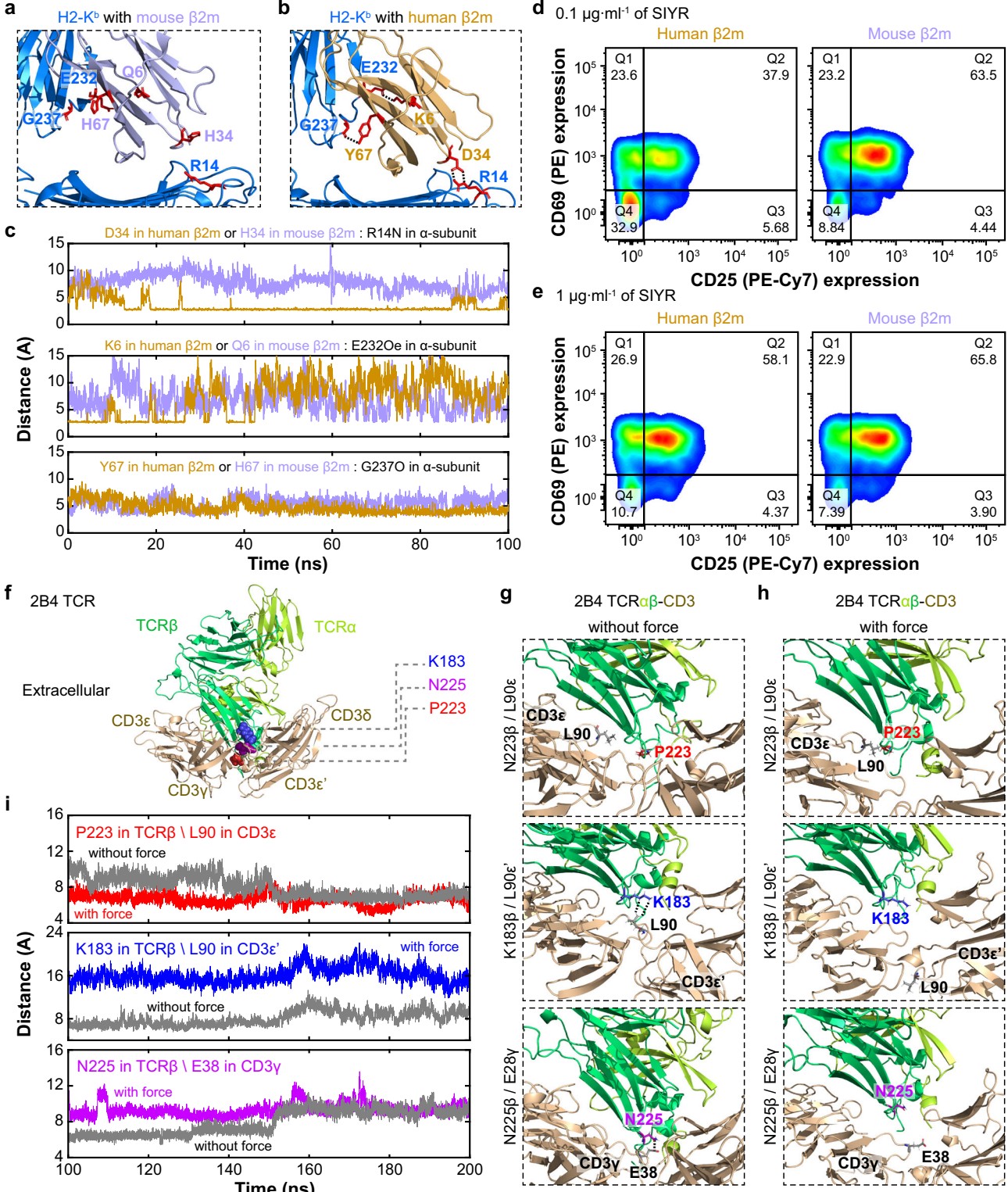

**Fig. 7 | Model validation by mutagenesis. a–c** Comparison of structures (**a**, **b**) or noncovalent contacts (**c**) of interactions of the H2-K$^b$ $\alpha_1\alpha_2$ (blue) with mouse (**a**, **c**) and human (**b**, **c**) $\beta_2$m (purple for mouse and human for orange, respectively). The structures in (**a**, **b**) are depicted by ribbon diagram using snapshots from SMD stimulations (initials modeled using 1G6R and 2BNR) with side-chains of the interacting residues shown by sticks (red). Simulated time courses of distances between the interacting H2-K$^b$ $\alpha$ chain residues and $\beta_2$m residues are plotted in **c**, showing shorter distances with the human $\beta_2$m and longer distances with the mouse $\beta_2$m. **d**, **e** Comparison of potencies to activate naive CD8$^+$ 2 C T cells by hybrid (left column) and WT (right column) R4:H2-K$^b$ at 0.1 µg/ml (**d**) or 1 µg/ml (**e**) concentration for 72 h. T-cell activation was assayed by flow cytometric analysis of

upregulation of surface markers CD69 (y axis) and CD25 (x axis) using PE-conjugated anti-CD69 and PE-cy7-conjugated anti-CD25 antibodies. **f** Structure of 2B4 TCRαβ showing the locations of residues N222-P223, G182-K183, and N225-H226 on Cβ domain with CD3 complex (6JXR). **g**, **h** Comparison of interactions of P223 (1st row), K183 (2nd row), and N225 (3rd row) with the corresponding CD3 residues in the absence (**g**) and presence (**h**) of force using MD simulations (initial build upon 6JXR). **i** Simulated time courses of distances between Cβ P223 and CD3ε L90 (1st row), Cβ K183 and CD3ε' L90 (2nd row), as well as Cβ N225 and CD3γ E38 (3rd row) in the absence (gray) and presence (colored) of force. Source data are provided as a Source Data file.

probes[8,12,46–48]. Also, except for two papers that failed to observe catch bonds in the soluble 1G4 TCRαβ ECD using a flow chamber[49,50], extensive data in 9 papers from four laboratories[2–4,6,8,10,13,25,26] plus additional data presented in this study have demonstrated catch bonds in 12 TCRs (including the full 1G4 αβTCR complex on the cell membrane) using BFP and OT. These experiments prompted us to develop two mathematical models for TCR catch bonds following the 1D formulation of Guo et al.[19], one with class I and the other with class II pMHC, based on Kramer's kinetic theory and accounted for the 3D coarse-grained structures, molecular elasticity, and conformational changes of the TCR–pMHC-I/II complexes. Previously, several models have been developed to describe catch-slip bonds of intermolecular interactions, including selectins–ligands[24,51–53], platelet glycoprotein Ibα–von Willebrand factor[54], FimH–mannose[51,55], sulfatase Sulf1–glycosaminoglycan[56], integrins–ligands[57,58], myosin–actin[59], cadherin–catenin/F-actin[60,61], vinculin–actin[62], and talin–actin[63] interactions. Except for the sliding-rebinding model, which is based on force-induced conformational changes in P-selectin–ligand observed from SMD simulations[53], none of these models have included any specific structural considerations of the interacting molecules. Instead, these models are based on a generic physical picture of dissociation along two pathways from either one or two bound states in a 2D energy landscape that is tilted by force[18,64]. Except for the two-pathway model tested here, which has 4 parameters[24], all other models have 5–10 parameters; therefore, over-fitting is a concern for applying them to some of the datasets analyzed here. Although the two-pathway model is capable of fitting the TCR–pMHC catch-slip bond data well, as shown previously[4] and tested more extensively by much larger datasets here (Supplementary Fig. 1 and Fig. 5d), the four best-fit model parameters correlate with neither T-cell function nor the catch-bond intensity for either class I or class II system (Supplementary Fig. 6), hence informing no insights on biology or bond profile. Consequently, it cannot distinguish the class I and class II systems because the parameters of the generic two-pathway model have nothing to do with the differential structures of the class I and II systems. This comparison highlights the utility and usefulness of our models and suggests opportunities for developing system-specific models based on the structures, elasticity, conformational changes, etc., to better describe the catch bonds of the above listed interactions.

Force-induced conformational changes of TCR–pMHC-I complexes have been observed or suggested by single-molecule experiments and SMD simulations[4,6]. Parameterizing these conformational changes by the number of unfolded amino acids $n^*$ and the bonding interface tilting angle $\theta$ in the class I model allows us to explain mechanistically and quantitatively the TCR–pMHC-I catch-slip and slip-only bonds. Indeed, the criteria for catch-slip bond are $n^* > 0$ and $\theta > 0$; the greater their values the more pronounced the catch bond. Importantly, the validity of the class I model has been supported by its capability to fit almost all force-lifetime datasets published to date plus one dataset presented here, and by the correlation between the best-fit model parameters and the available biological activity data induced by the TCR–pMHC-I interactions.

By comparison, the respective ranges of $n^*$ and $\theta$ for the class II model are smaller, consistent with the sturdier structure of the pMHC-II molecule[21]. Mutagenesis studies and MD simulations in the present work have supported the hypothesis of force-induced conformational changes in the TCR structure. Our class II model has also been tested by all published datasets plus four datasets presented here, and their best-fit parameters also correlate well with the biological activities induced by the TCR–pMHC-II interactions. Furthermore, the validity of models of both classes has been supported by the findings that the best-fit model parameter $\delta_0^*$ correlates with the catch-bond intensity $I$ and that the $\delta_0^*$ vs $I$ and $\delta_0^*$ vs $n^*$ plots have more appropriate slopes and $R^2$ values when the model matches than mismatches the data.

We should note that since TCR–pMHC unbinding is assumed to be a spatially continuous and temporally instantaneous process, all structural parameters determined here represent mean field values, and they were evaluated by fitting the mean bond lifetime vs force data. However, individual bond dissociation events are inherently stochastic; and as such, need not be deterministically mapped onto any specific conformational changes in a one-on-one fashion. Instead, any particular bond type and parameter sets are related on the average sense. Future studies are required to extend the current framework to relate more detailed structural changes and bond lifetime distributions, e.g., to account for more sequential partial unfolding events prior to transition state as suggested by experiments[4,5,65].

A strength of our agent-based models lies in their ability to incorporate many different ideas and knowledge into a simple 1D formulation. This simplicity facilitates model application to both class I and II experimental systems, enables quantitative interpretation of TCR–pMHC bond lifetime vs force profiles, expresses biological functions by biophysical measurements, and suggests structural mechanisms of how the TCR mechanotransduction machinery might work. However, the 1D simplification is also a weakness because theoretically these models can only describe single-step dissociation by entropic conformational fluctuation in the low-force regime from a single-state along a single-dissociation path, implicitly assuming that there is only a single energy barrier. Although some catch-slip and slip-only bonds can be described by such simple models[3], more complicated TCR–pMHC bonds has been reported. These are evidenced by the multi-exponential bond lifetime distributions at constant forces, which have been fitted by data-driven multi-state, multi-pathway models[10]. To address this weakness, future studies may extend the present 1D model to 2D, e.g., by combining Eqs. (2) and (4), to enable proper description of multi-exponential bond survival probabilities.

We introduced the catch-bond intensity $I$ as a dimensionless scaled metric for the bond lifetime vs force curve and generated four model parameters that describe the curve's geometric features. Upon analyzing all 49 catch-slip and slip-only bond profiles published to date by four independent laboratories[2–4,6,8,10,13,25,26] plus 6 additional ones reported here, we found that these quantities do a better job to predict TCR function than any other quantities. This finding strongly suggests the relevance of catch bond of TCR to its unique properties, e.g., sensitivity, specificity, ability to discriminate self vs nonself, etc. For example, it may explain how force amplifies TCR signaling and antigen discrimination, because $I$ is defined by a force curve and $n^*$ and $\theta$ only predict signaling when they assume none-zero values at $F > 0$. It should be noted that despite the comparable force ranges, highly variable lifetimes have been observed for different TCR systems interacting with different pMHCs (e.g., WT vs hybrid H2-K^b). Even the same TCR–pMHC interactions could display different bond lifetimes in the absolute scale, depending on the cells on which the TCR is expressed. The power for the catch-bond intensity $I$ to predict TCR signaling and discriminate antigen may lie in the ability of this dimensionless number to capture different bond lifetime patterns in a relative scale.

A recent study showed surprising features of reversed-polarity of TRBV TCRs such that interactions of $NP_{366}$:H-2D^bD227K to TCRs B13.C1 and B17.C1 induced T-cell signaling, whereas interactions of the same pMHC to B17.R1 and B17.R2 TCRs did not[13]. Despite that the former two TCRs formed catch-slip bonds with $NP_{366}$:H-2D^bD227K and the latter two TCRs formed slip-only bonds, the authors suggested that the signaling capability of the B13.C1 and B17.C1 TCRs could not be attributed to their force-prolonged bond lifetimes because the B17.C1 TCR–H-2D^bD227K bond was shorter-lived than the B17.R2 TCR–$NP_{366}$:H-2D^bD227K bond across the entire force range tested. Even at 9.4 pN, which was $F_{opt}$ for the former with a $t_{peak} = 0.61$ s, the latter lived 2.48 s on average, and the longest lifetime of the latter was $t_0 = 2.83$ s occurred at zero force[13]. The authors hypothesized that the TCR–pMHC docking orientation, which was "canonical" for the B13.C1 and B17.C1 TCRs but

"reversed" for the B17.R1 and B17.R2 TCRs, underlain the signaling outcomes by directing the position of Lck relative to the CD3. However, we suggest that even without knowing the docking orientation, our model parameters are capable of determining the signaling outcomes. Indeed, our analysis correctly maps the data of the B13.C1 and B17.C1 TCRs onto the high peptide potency region and the data of the B17.R1 and B17.R2 TCRs onto the low peptide potency region of the $\delta_0^*$ vs $I$ (Fig. 4c), $\delta_0^*$ vs $\theta$ (Fig. 4e), and $\theta$ vs $d_{\alpha 3}$ (Fig. 4f) phase diagrams. Thus, by mechanistically modeling the effect of force on bond dissociation, TCR signaling and antigen discrimination can be predicted by the model parameters.

The success in our model applications indicate that the conformational changes assumed in the models may be important to the TCR triggering, thereby suggesting testable hypotheses for future studies designed to investigate the inner workings of the TCR mechanotransduction machinery, e.g., to extend and/or revise models regarding how TCR signaling is triggered. Some TCR triggering conceptual models involve conformational changes and/or catch-bond formation[66–68]. Our structure-based biophysical models relate catch and slip bonds to TCR−pMHC conformational changes. For the class I model, the parameterized structural changes include force-induced disruption of the MHC $\alpha_1\alpha_2$−$\beta_{2m}$ interdomain bond, partial unfolding of the $\alpha_1\alpha_2$−$\alpha_3$ joint, tilting of the TCR−pMHC bonding interface, and partial unfolding of the V$\alpha$-C$\alpha$ and V$\beta$-C$\beta$ joints. For the class II model, these are primarily limited to the force-induced partial unfolding of the MHC-II $\alpha_1$-$\alpha_2$ and $\beta_1$-$\beta_2$ joints as well as the V$\alpha$-C$\alpha$ and V$\beta$-C$\beta$ joints. Besides these, one additional conformational change observed in the SMD simulations of TCR$\alpha\beta$−pMHC dissociation is unfolding of the connecting peptides between the TCR$\alpha\beta$ ECD and transmembrane domain[6]. We chose not to include this conformational change in our models because such unfolding would likely be prevented by the interaction of the C$\alpha\beta$ with the CD3 subunits. Consistent with this assumption, the experimental data used for model fitting to evaluate conformational change parameters ($n^*$ and $\theta$) are those of pMHC bonds with TCR−CD3 complexes on the cell membrane that includes the TCR$\alpha\beta$ ligand binding subunits and the CD3 signaling subunits (except for the N15 TCR$\alpha\beta$ case which is soluble ECD only). Indeed, our previous work found that catch bonds of purified TCR$\alpha\beta$ were altered from those of cell surface TCR interacting with the same pMHCs[10,25], which is reflected by their changed model parameters (Supplementary Fig. 12). As such, the TCR$\alpha\beta$ conformational changes predicted by our models provide a constraint for possible CD3 conformational changes in the TCR−CD3 complex to be considered in future TCR triggering models. Indeed, our data on WT and MT 2B4 TCR−K5:I-E$^k$ interactions indicate the importance of the TCR$\alpha\beta$−CD3 cis-interaction on catch-bond formation of the TCR−pMHC trans-interaction.

Another constraint to be considered by future studies is that imposed by the coreceptor CD4 and CD8, as co-ligation of the coreceptor prolongs bond lifetimes, amplifies catch bonds, and may even changes slip-only bonds to catch-slip bonds[7,8,10,13,26]. Future studies should also consider how to extend the current models to pre-TCR catch and slip bonds with a broad range of ligands[65,69]. Instead of the TCR$\alpha$, the pre-TCR uses the TCR$\beta$ chain to dimerize with a common pre-T$\alpha$ chain, which lacks the variable domain (hence no V$\alpha$-C$\alpha$ hinge). Without extension, even if our models are still able to fit the data of pre-TCR−ligand bonds or data of TCR−pMHC bonds where the TCR$\beta$ F−G loop was deleted or bound by an anti-TCR antibody[4], the best-fit model parameters may not correspond to the conformational changes of these molecular complexes which are likely different from the conformational changes in the TCR−pMHC bonds with intact F−G loop.

An objective of this work is to explore the extent to which 1D models can describe experimental data with a minimal set of meaningful parameters. Our parameters consider coarse-grained structural features and relate catch and slip bonds to specific force-induced conformational changes of the TCR−pMHC complex. This approach should be extendable to the modeling of other receptor−ligand systems of different structural features yet also form catch and slip bonds, such as selectins[15,70,71], integrins[57,58,72–75], cadherin[76], Fc$\gamma$ receptor[77], notch receptor[78], platelet glycoprotein Ib$\alpha$[54,79], FimH[55], actin with myosin[59], actin with actin[80,81], cadherin−catenin complex with actin[60], vinculin with actin[62], talin with actin[63], and microtubule with kinetochore particle[82].

Our models allow us to develop working hypotheses regarding how T-cell function is regulated through structural modulations of catch and slip bonds. For example, in this study we validated a prediction of the class I model that strengthening of the $\alpha_1\alpha_2$−$\beta_{2m}$ interdomain bond would weaken the TCR−pMHC catch bond, which would in turn reduce T-cell activation. This prediction has also been supported by our published data that somatic mutations in HLA-A2 found in some cancer patients impair TCR−pHLA-A2 catch bonds, which may explain the suppressed anti-tumor T-cell immunity[6]. More interestingly, our models pave the way for engineering of TCR function for tumor immunotherapy by modulating the TCR catch and slip bonds through alteration of its structures. For example, we have shown a mutation that weaken the TCR$\alpha\beta$−CD3 ECD cis-interaction under force amplifies TCR catch bond and enhances the T-cell effector function, which suggests a strategy that may be more advantageous compared to mutations at the pMHC docking interface because mutations at the C$\beta$−CD3 ECD binding interface are not expected to alter the TCR specificity but the same mutation may be effective to different TCRs specific for different tumor antigens. By comparison, mutations at the TCR binding interface may be applicable to a specific pMHC only and may be riskier in terms of cross-reactivity to self-pMHCs. Thus, rational design guided by catch-bond models may provide additional TCR engineering strategies that warrant future studies.

## Methods

All experiments in this study were conducted with compliance to the Institutional Review Board of Georgia Institute of Technology and Emory University IACUC-approved protocol.

### Cells and proteins

Naive CD8$^+$ T cells were purified by negative selection from spleens of 2C transgenic mice housed in the animal facility of Emory University following a protocol approved by the Institute Animal Care and Use Committee of Emory University as described[2]. Briefly, C57BL/6J mice (Jackson Laboratories, strain #000664) were used to generate 2 C TCR transgenic mice. We housed 2C transgenic mice in plastic cages with disposable bedding under standard conditions, including a 12-h dark/light cycle, 40–60% humidity, and temperatures ranging from 18 to 23 °C, with free access to food and water. We used male or female mice aged 6–8 weeks to purify primary T cells from the spleen. After the 2C transgenic mice were sacrificed via CO2 induction with a fill rate of 1.7–3.9 L/min with 30–70% of the chamber filled per minute, the spleen was harvested to isolate T cells. After being packed into the red biohazard bag, the bodies of the sacrificed mice were stored in the Necropsy room's −20 °C freezer. For each round of BFP experiment, we utilized primary T cells that were purified from the spleen of mice as required. Mouse 58$^{-/-}$ T-cell hybridoma cells[83] expressing mouse CD3 but not TCR$\alpha\beta$ were a generous gift from Dr. Bernard Malissen (Centre d'immunologie de Marseille-Luminy, France). WT or MT mouse 2B4 TCR were re-expressed on 58$^{-/-}$ cells through retroviral transduction, which were cultured as described[38]. The transduced cells were stained with PE anti-mouse CD3$\epsilon$ (clone 145-2C11 or 2C11, eBioscience, 12-0031-82, 1:20) and allophycocyanin (APC)-conjugated anti-TCR$\beta$ (clone H57-597 or H57, eBioscience, 17-5961-82, 1:20) mAbs and sorted for dual expression of CD3 and TCR. The sorted cells were expanded for 6 days and quantified for TCR and CD3$\epsilon$ expression.

C-terminally biotinylated WT and $\beta_2$m swapping hybrid H2-K$^b$ presenting the R4 peptide (SIYRYYGL) were from the National Institutes of Health Tetramer Core Facility at Emory University. To prevent CD8 binding, the MT H2-K$^b$α3A2 (with the mouse α3 domain swapped to that of the HLA-A2) was used. Inclusion bodies for I-E$^k$ α (with C-terminal biotinylation sequence) and β chains were produced in One Shot™ *E. coli* BL21 (DE3), refolded with K5 peptide (ANER-ADLIAYFKAATKF), and purified as described previously[84].

Human red blood cells (RBCs) for BFP experiments were isolated from the whole blood of healthy volunteers according to a protocol approved by the Institutional Review Board of Georgia Institute of Technology as described[2]. In total, 20–50 μl of human RBCs were isolated from blood of healthy donors (healthy male and female (not pregnant) adult donors aged 20–40 who weigh at least 110 pounds) according to a protocol approved by the Institutional Review Board of Georgia Institute of Technology with informed consent from the donor. After washing with carbonate/biocarbonate buffer (80 mM Na$_2$CO$_3$ 126 mM NaHCO$_3$ in diH$_2$O) twice by centrifuge (2000 rpm for 2 min), RBCs were mixed with biotin-PEG3500-NHS (Jenkem Technology) at pH of 8.5 for 30 min at room temperature. After biotinylated RBCs were washed twice with N2 buffer (265.2 mM KCl, 38.8 mM NaCl, 0.94 mM KH2PO4, 4.74 mM Na$_2$HPO$_4$, and 27 mM sucrose; pH 7.2 at 588 mOsm), biotinylated RBCs were mixed with nystatin for 30 min on ice. After washing twice with N2 buffer, Nystatin-treated biotinylated RBCs were then resuspended in 200 μl of N2 buffer, and stored at 4 °C for BFP experiments.

### BFP bond lifetime measurement

A previously described BFP force-clamp assay was used to measure TCR–pMHC bond lifetimes in a range of constant forces at room temperature[2]. Briefly, pMHC was coated onto streptavidin-conjugated glass beads via biotin-streptavidin coupling. A pMHC-coupled bead was attached to a biotinylated RBC aspirated on a glass micropipette to form a force probe to test binding with a primary T cell or hybridoma expressing the specific WT or MT TCR in repetitive cycles. In each cycle, the cell was driven by a piezo translator controlled by a computer program to approach and briefly (~0.1 s) contact the probe bead with a small impingement force (~10 pN) to allow bond formation, followed by retraction of the cell at a force loading rate of 1000 pN/s. If a bond was detected at a preset tension level, the force was clamped until spontaneous bond dissociation. Bond lifetime was measured as the duration of force clamp. To ensure most adhesion events were mediated by single molecular bonds, the adhesion was controlled to be infrequent (≤20%)[85]. Bond lifetimes were measured at forces ranging from 2 to 30 pN, pooled, and binned into >7 force bins (>50 measurements per bin) to reduce system errors and presented as mean lifetime and standard error of the mean (SEM). A previously described thermal fluctuation assay was used to measure bond lifetime at zero force[86]. Here, instead of retracting the T cell to apply a tensile force as in the force-clamp assay, the retraction stopped when the contact force disappears and the TCR and the pMHC were then allowed to interact via thermal fluctuation of the probe bead. Bond association and dissociation were identified from reduction and resumption of thermal fluctuation of the bead position. Individual lifetimes were measured as the duration from fluctuation reduction to resumption. Measurements (>10 cell-bead pairs from three independent experiments) were recorded using Labview 2016.

### In vitro T-cell activation

Upregulation of CD25 and CD69 on naive 2C T cells were assayed using 96-well plates pre-coated with WT or hybrid SIYR:H2-K$^b$ at 0.1 μg/mL or 1 μg/mL concentrations for 1 h at 37 °C. Upon addition of naive 2C T cells at 1 million per well, the plates were incubated at 37 °C for 72 h. Cells were harvested and analyzed for fluorescence staining. For checking CD8 expression, APC-anti-mouse CD8 (clone 53-6.7, BD

Biosciences, 553035, 1:20) was used. For measuring TCR activation, PE anti-CD69 (clone H1.2F3, BD Biosciences, 553237, 1:20) and PE-cy7-anti-CD25 (clone PC61, BD Biosciences, 552880, 1:20) were used and measured by flow cytometry (BD FACSAria). Flow data were analyzed by software (FACSDiva v9 and Flowjo v10). All cell lines were first gated on FSC/SSC and gating was based on the expression or coating of the molecule of interest.

### Molecular dynamics simulations

**Molecular modeling of the hybrid H2-K$^b$.** Two complex models of human $\beta_2$m and H2-K$^b$ were built based on the crystal structure of mouse $\beta_2$m and H2-K$^b$. Because of the high sequence identity (68%) and high structural similarity (backbone RMSD < 1 Å) between the human and mouse $\beta_2$m, we made in silico mutation to replace mouse $\beta_2$m residues by those of human $\beta_2$m to the WT H2-K$^b$ (PDB ID: 1G6R), or replace the entire mouse $\beta_2$m by the human $\beta_2$m in the HLA-A2 (PDB ID: 2BNR).

**Stability comparison between hybrid and WT H2-K$^b$ by conventional MD.** Upon adding hydrogen atoms and counter ions (~150 mM NaCl) and solvating the structures in rectangular water boxes (>16 Å from the box edges and protein) by the VMD software package, we obtained two solvated systems—one for the hybrid and the other for WT H2-K$^b$—with dimension of ~92 × 82 × 97 Å$^3$. Both systems were first equilibrated with three steps: (1) 10,000 steps energy minimization and 4-ns equilibration simulations under 1-fs timestep with heavy atoms constrained (except difference residues between mouse $\beta_2$m and human $\beta_2$m); (2) 4-ns equilibration simulations under 1-fs timestep with backbone atoms of proteins constrained; (3) 10-ns equilibration simulation under 1-fs timestep without constrains. Subsequently, the production simulations last ~100 ns with 2-fs timesteps under rigid bond algorithms to relax the models. Energy minimizations and MD simulations were performed with NAMD2 using the CHARM36m force field for proteins under periodic boundary conditions. Temperature was maintained at 310 K with Langevin dynamics and pressure was controlled at 1 atm with the Nosé–Hoover Langevin piston method. Particle Ewald Mesh summation was used for electrostatic calculation and a 12-Å cutoff was used for short-range non-bounded interactions.

**Modeling and simulation of the TCR–CD3 ectodomain interaction.** All simulations were based on the recently published cryoEM structure of a human TCR–CD3 complex (PDB ID: 6JXR)[43], which shares the same Cαβ, CD3γε, and CD3δε' with the mouse 2B4 Vαβ and human LC13 Cαβ hybrid TCR used in our experiments. The structure was transmembrane domain truncated, end ACE/NME capped, and missing residues repaired[87] to form a complete CD3δε'–TCR–CD3γε trimeric ECD complex. Unit cells were built to enclose the molecular systems to be simulated in a physiologically appropriate and thermodynamically favorable state. The initial structures were oriented and centered within optimized orthorhombic cells, which were subsequently solvated using the TIP3P water model[88], counter-balanced using sodium ions, and ionic strength tuned to ~150 mM with sodium chloride. To achieve a thermodynamically favorable initial state, the unit cell was energy minimized, followed by two equilibration cycles under NTV, then NPT ensembles with the heavy-atom restraints. The systems were then ready for subsequent equilibration and production runs with/without external force applied using GROMACS (version 2019.6)[89–91] under the AMBER99SB*-ILDNP force field[92].

Conventional molecular dynamics (CMD, without force) simulations were performed by letting initials freely evolve without any constraints. While in steered molecular dynamics (SMD, with force) simulations, the external constant forces with constant magnitudes (175 pN) along a fixed direction (z axis) were added to the ECD initials. The N-terminus of the TCR α chain was pulled; in the meantime, the C-termini of CD3ε chains were fixed to mimic the anchor effect of their

transmembrane domains. Every simulated trajectory consists of a 100-ns equilibration and a 100-ns production stage. The conformations per 2 ps during the production stage were analyzed to obtain the center-of-mass distances between interested residues. The snapshots per 400 ps during the same period were extracted for visual comparisons.

### Modeling and curve of fitting of TCR−pMHC force-dependent bond lifetime profiles

The model developments, characterization, and validation are described in the main text with more details in the Supplementary Model Derivations. Initial states and force-free end-to-end distance of TCR−pMHC complex were identified using PyMol 2.3. Model fitting to experimental data was done by nonlinear curve-fitting in the least-squares sense using the Levenberg-Marquardt algorithm (MATLAB built-in function). Briefly, the best-fitting parameter set was derived by fitting model to mean value of bond lifetime vs force profile, and SE of fitting was calculated by independent fitting to mean + SEM and mean − SEM of bond lifetime vs force data, and found that the parameters fitted to the mean of bond lifetime were robust and in the range parameters ±SEM (as detailed in Supplementary Model Derivation, A.4. Model applications, curve-fitting strategies, and biological relevance). Clustering analysis using Lloyd's algorithm was done by MATLAB built-in function (all analyses were done using MATLAB 2020b). All published experimental bond lifetime vs force data were measured at room temperature as reported in refs. [2–4,6,8,10,13,25,26] and the TCR−pMHC constructs used by these studies were described in the footnotes of Supplementary Tables 3, 4, and 6.

### Statistics and reproducibility

Each scatter in bond lifetime vs force profiles measured in this study as well as previously reported indicates mean ± SEM calculated from at least >20 individual lifetime data per each force bin (for BFP, > 50 individual lifetime data per each force bin). In statistical analyses, a linear fitting was applied using the least-squares method, and (paired or unpaired) $t$ test with one- or two-sided was performed using MATLAB 2020b. BFP experiments were performed at least three times with random selection and blinding. No statistical method was used to predetermine sample size. Only clear binding events with high signal-to-noise were used for analysis of bond lifetime. For flow cytometry, the cells were randomly measured by the instrument. No statistical method was used to predetermine sample size. No data were excluded from the analyses after gating. Multiple independent MD simulations were performed with the maximum sample size (each 50,000 frames from the production phase for data analysis) to ensure that the differences between the results of different simulations systems are statistically significant according to SEM. The data from the equilibration phase were excluded, where the molecules were reshaping under the influences of domain-swapping or external constraints to bring the system to the desired conditions. The stabilized structures and dynamics from the production phase were valid representations of experimental results. During the simulation, initial states (velocities for each atom) were randomly sampled from the Maxwell-Boltzmann distribution for each independent run. All methods were not relevant to blinding and not biased by the investigators.

### Reporting summary

Further information on research design is available in the Nature Portfolio Reporting Summary linked to this article.

## Data availability

All data supporting the findings of this study are included in the article and Supplementary Information or from the corresponding author upon request. Previously published bond lifetime data[2–4,6,8,10,13,25] re-analyzed for model fitting are summarized and deposited in Github (https://github.com/Chengzhulab/Catch-bond-model_TCR-pMHC) or

available at Zenodo[93]. PDB structures were used to either apply MD simulation or identify the structural variabilities (end-to-end-/inter-distance of the bound state and the angle between domains) (2C TCR complexed with H2-K$^b$ (PDB codes 2CKB, 1MWA, and 1G6R) and H2-L$^d$m31 (2E7L), 1G4 TCR complexed with HLA-A2 (2BNR and 2BNQ). P14 TCR (5M00), NP1-B17 TCR complexed with H2-D$^b$ (5SWZ), E8 TCR with HLA-DR (2IAM, 2IAN), 2B4 with I-E$^k$ (6BGA, 3QIB), and TCR−CD3 complex (6JXR). Source data are provided with this paper.

## Code availability

All codes for three models used in this study are summarized and deposited in Github (https://github.com/Chengzhulab/Catch-bond-model_TCR-pMHC) or available at Zenodo[93].

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

## Acknowledgements

We thank Jinsung Hong, Peng Wu, and Tongtong Zhang for sharing their published BFP experimental data for re-analysis and model fitting. We also thank Brian Evavold (Emory University) for providing 2CT cells and Bernard Malissen (Centre d'immunologie de Marseille-Luminy, France) for sharing 58$^{-/-}$ hybridoma cells. We acknowledge the National Institutes of Health Tetramer Core Facility at Emory University for providing the pMHC molecules. The MD simulations were supported by an NSF award (MCA08X014) in advanced computing infrastructure for U.S. and performed in the Extreme Science and Engineering Discovery Environment (XSEDE). This work was supported by a Postdoctoral Fellowship from the National Research Foundation of South Korea (2021R1A6A3A03038382 to H.-K.C.), by a National Natural Science Foundation of China grant (31971237 to W.C.), and by a National Natural Science Foundation of China grant (32090044 to J.L.). This work was also supported by National Institutes of Health grants (U01CA250040 to C.Z., U01CA214354 and R01CA243486 to C.Z. and M.K., and R01GM124489 to M.K. and C.Z.).

## Author contributions

H.-K.C. and C.Z. conceived the project and developed the models. P.C., C.G., B.L., and A.N. performed experiments. P.C., Y.Z., and J.L. performed MD simulations. H.-K.C., K.L., and M.N.R. provided data for re-analysis prior to their publication. H.-K.C. performed model analysis and data fitting. H.-K.C., K.L., and C.Z. made key observations and conclusions from analyses. W.C., M.K., J.L., and C.Z. supported the project. H.-K.C. and C.Z. prepared the manuscript with input from other authors.

## Competing interests

The authors declare no competing interests.
