## [Peer Review File · Nature Communications]

Catch bond models may explain how force amplifies TCR signaling and antigen discriminationEditorial Note: Parts of this Peer Review File have been redacted as indicated to maintain the confidentiality of unpublished data.

REVIEWER COMMENTS

Reviewer #1 (Remarks to the Author):

Detailed comments on the ms. titled "Catch bond models explain how force amplifies TCR signaling and antigen discrimination" by Choi et al. (for submission to the authors)

I. Summary

Choi et al. lay out a mathematical approach to predict structural changes as they may occur within peptide presenting MHC class I and class molecules and their cognate $\alpha\beta$ TCRs upon complex formation within the confines of the immunological synapse. The work is inseparably tied to the concept of catch-slip bonds, as they have been first described for stimulatory TCR-peptide/MHC (pMHC) interactions in 2014 by the corresponding author of this ms.. Based on Kramer's kinetic rate theory the authors formulate a rationale, in which they assume a semi-rigid body nature of the TCR-pMHC complex (which was earlier predicted by the corresponding author via steered molecular dynamics (SMD) simulations) to calculate extensions of more flexible inter-domain linkers, dissociations of interdomain contacts, as well as inter-domain rotations under mechanical force. Such force-dependent structural changes are ultimately projected to create binding conditions under which catch-slip bonds take effect: force-directed tilting of the interaction surfaces between TCR- and pMHC. This behavior then serves as an explanation for what the authors term "catchiness". To validate their theoretical leads the authors correlate calculated catch-slip bond behavior with force-bond lifetime plots as they have resulted from carrying out two-dimensional BFP-based force-response assays. Given observed similarities between predicted and measured "catchiness", the authors present a mechanistic explanation of how the phenomenon of catch-slip bonds may allow T-cells to discriminate antigenic from endogenous pMHCs and also how TCRs are triggered upon extracellular ligand engagement to transduce signals across the plasma membrane.

II. Reviewer's concerns

Explaining how T-cell recognition proceeds on a molecular level is without much doubt a task of colossal proportions. As a reader and reviewer of the ms. I find the presented wealth of intriguing reasoning refreshing. Most importantly, this reasoning leads to testable hypotheses, which have the potential to transform our understanding of T cell recognition with palpable implications for clinical translation, if the validity of these hypotheses holds firm after rigorous wet lab testing.

Unfortunately, the authors refrain from providing direct evidence for the proposed structural changes, which I consider the central element of their line of thinking. Instead, they base their data validation entirely on previously reported experiments or simulations, which do not qualify as direct proof. More specifically, the authors refer to (i) SMD simulations in combination with (ii) experimental findings derived either from thermo-fluctuation / force-lifetime assays (which are indirect in nature as they require a number of yet unverified assumptions, e.g. single molecule detection, lack of cellular changes after repetitive testing on the same T cell), or from (iii) the use of magnetic tweezer-based force cycles, which give rise to a noisy readout. The latter approach has so far lacked statistical corroboration, and solid conclusions on force-induced structural changes are hence not yet supported. Looking at all the evidence provided, I am not convinced of structural changes within the TCR-pMHC complex upon synaptic bond formation, as other explanations are still possible. My opinion would change if I were offered clear evidence, such as (single molecule) life cell microscopy reporting on the deformation of synaptic TCR-pMHC complexes (or TCR or pMHC). For example, experiments involving intramolecular FRET within pMHC or the TCR could in real time and most directly confirm the kind of structural changes put forward in the ms..

Yet this direct experimental evidence is so urgently needed to resolve many of the controversies in the field, which is - unlike what the authors wish to portray - divided into a “catch-slip bond-” and “slip-bond-camp“ and a large group of scientists who do not know whom to believe. What adds to my concerns is that fact that the authors willfully ignore findings by others that challenge the core of the catch-slip bond hypothesis or the relevance of catch-slip bonds for TCR-proximal signaling. To name 2 of many examples, there is no mentioning of the (i) bead-based experiments done in laminar flow by Philippe Robert and colleagues, who failed altogether to directly observe any catch-bond behavior of 5 well-studied TCR-pMHC pairs (PMID: 31315981). (ii) The authors do not cite the force measurements by Schutz and his team on single TCRs engaging pMHCs or anti-TCR scFVs (PMID: 33947864). These single molecule measurements were conducted within the confines of the area of contact between T cells and stimulatory planar glass-supported lipid bilayers, and measured forces amounted to less than 2pN for TCRs bound to bilayer-attached anti-TCR scFV, i.e. 10-times lower forces than the forces that catch bonds are reported by Zhu and colleagues to require in order to exert their effect on TCR-pMHC lifetimes. Of note, the force loading rate measured by the Schutz group amounted to about 1.5 pN/s which renders it highly unlikely to arrive at 10-15 pN forces with the rather short pMHC-TCR lifetimes measured by means of a BFP (PMID: 24725404), single dye tracing (PMID: 23840928) or FRET-based recordings (PMID: 20164930).

While these two studies do not necessarily rule out the existence of TCR-pMHC catch bonds or their relevance for antigen recognition (also in view of much higher force values measured by Salaita's team, PMID: 27140637), they certainly need to be reckoned with within the context of his ms.. In essence, such findings demand that TCR-pMHC catch bonds as well as the structural changes that may cause them to take effect (as elaborated in this ms.) need to be demonstrated most directly (and not indirectly which would leave room for alternative explanations) as a means to establish them beyond doubt as a principle underlying T cell antigen recognition and ligand discrimination.

In the absence of such demonstration and a more balanced description of the current state of findings the ms. runs the risk of being perceived as “prediction in hindsight”, in particular since its findings are predominantly based on data gathered by the lab of the corresponding author of this ms.. Also, the ms. would certainly gain in credibility by establishing data transparency. This may include providing the means to reproduce the mathematical proceedings and making unprocessed primary data (e.g. BFP-based experiments, magnetic tweezer experiments) available for download.

Taken together these considerations (lack of direct and clear evidence for structural changes, overstatements regarding the implications of the findings/results for mechanisms underlying TCR-based pMHC discrimination and signaling, highly unbalanced description of the state of research) prevent me from recommending publication of this study in Nature Communications. Given the emphasis on mathematical modeling, and the effort it takes the average life scientist to understand the content, the ms. may be more suitable for a more specialized journal targeting physicists or life scientists with an extensive physics background.

III. Specific points

1. The ms. is well structured. Overall the figures are clear. See last point (Suggestions).
2. Experimental details (temperatures, constructs) should be indicated more clearly for all listed experiments. I found myself too often looking these data up in the literature.
3. Confidence intervals or error bars are missing in the following figures: Fig. 1c, e, g, Fig. 2c, d, f, e, Fig. 3b, Fig. 4c.
4. How do the authors arrive at the demarcation separating Catch-only, Catch to slip and slip-only in Fig. 1c lower panel?
5. Some of the wording in the Supplementary Information section needs editing.

6. Specific examples of wording which I find problematic

- Title:

“Catch bond models explain how force amplifies TCR signaling and antigen discrimination”

The wording is too strong, as the work provides testable yet untested hypotheses. Changing the title to “Catch bond models may explain how force amplifies antigen discrimination” or similar would be more adequate.

- Abstract:

“Central to T cell biology, the T cell receptor (TCR) integrates forces in its triggering process upon interaction with peptide-major histocompatibility complex (pMHC)”

The wording is too strong and projects an unbalanced view of the current state of data.

“Phenotypically, forces elicit TCR catch-slip bonds with strong pMHCs but slip-only bonds with weak pMHCs. While such correlation is generally observed...”

The wording is too strong / unbalanced and does not capture the current state of ground truth finding.

“The extensive comparisons between theory and experiment allowed us to validate the models and identify specific conformational changes that control bond profiles, thereby providing structural insights into the inner workings of the TCR mechanosensing machinery and explaining why and how force amplifies TCR signaling and antigen discrimination.”

The wording is too strong and overstates the results.

- Introduction:

“Mechanical forces applied to TCR via engaged pMHC substantially increase antigen sensitivity and amplify antigen discrimination”

◇ A large part of the community will not agree with this strong statement.

“As a fundamental force-elicited characteristic, strong cognate pMHCs form catch-slip bonds with TCR where bond lifetimes increase with force until reaching a peak, and decrease as force increases further, whereas weak agonist and antagonist pMHCs form slip-only bonds with TCR where bond lifetimes decrease monotonically with increasing force.”

◇ There is plenty of evidence challenging these bold statements.

7. Suggestion

If targeting a broad life science readership is the aim, it may be helpful to provide a graphic explanation of the mathematical reasoning behind model development.

Reviewer #2 (Remarks to the Author):

Review

Choi et al.,

Catch bond models explain how force amplifies TCR signaling and antigen discrimination

In this manuscript, Choi and colleagues develop models for TCR–pMHC complex interactions, and in particular description of catch-slip bond and slip bond dynamics. The models are robustly tested across a number of datasets and experimental data is well used to develop key aspects of the models. The models do provide structural and mechanical insights into TCR interaction behaviors.

Overall, I commend the authors on the amount of model development, rigor (and rigorous testing) and

large-scale testing with multiple unique datasets. However, this reviewer is struggling to figure out how impactful and unique the new models and findings really are. Without direct comparison to the “next best” models it is extremely difficult to understand how much of an advance these models really are in terms of understanding unique aspects of TCR-pMHC mechanobiology. That said, there are cases where it appears clear that the models are an advance since the assumption is that published models would not be able to describe the data – however this is not well described by the authors in most cases. Likewise, the biological implications of the model findings are not robustly presented. For a general audience such as Nature Communications it would strengthen the work to make it very clear where the advances are and what new information is being gained. Along those lines, while the models are very elegant and insightful, the finding don’t bring the title to realization and explain how force amplifies TCR signaling and antigen discrimination. As such, this leaves a question for the Editor as to whether or not Nature Communication is the appropriate choice versus a more specialized journal.

This reviewer does not have any technical concerns from the model formulation, testing, and application. At times the model development is dense to work through and not particularly accessible to a general audience, but a deep dive into the model reveals its rigor. A few more minor comments: 1) The justification for the model needs on page 3 is not particularly compelling. It is also not clear why the authors consider catch-slip bond counterintuitive. When considering many of the key behaviors addressed in the manuscript they are quite intuitive, 2) as discussed above the implications of the model findings are not well developed, 3) it is not clear how the authors arrive at 48 datasets. Perhaps a table with all the datasets and what is being tested would be beneficial, and 4) (very minor), there are a number of typos and at times model terms are not defined on first mention.

Reviewer #3 (Remarks to the Author):

The authors propose a model of the catch and slip-bonds based on a one-dimensional (1D), single-well energy landscape, whose mathematical behaviour was developed in a previous publication (Guo et al.). In this article, the authors apply this model to describe TCR-pMHC complexes, exploit available data on the bond lifetime vs force relationship to assess how well such a model can explain it and can predict peptide potency, i.e. the strength of signalling upon TCR binding to a peptide.

In my opinion the work is valuable, the authors have for sure carried out an extensive and thorough analysis, however I am not convinced about how solid the main claim is, i.e. the claim that the model explains mechanistically and quantitatively catch-slip bonds in TCR-pMHC complexes. This claim is supported by the fit to data and by the correlation of 4 parameters (and not k_0) with peptide potency. My main concern (and I think the authors should try to explain this point very clearly) is the rationale by which one should expect a correlation of 4 parameters (and not k_0) with peptide potency and why instead the absence of this correlation can be taken as a way to select against a certain model. This correlation to potency (fig. 2f) should be represented by some scatter plot, giving a quantitation of correlation coefficient and its p-value. In addition, the fitting of models parameters to data is very unclear to me, it seems to me that the authors simply say ‘The curve-fitting strategies involve varying one parameter while keeping others constant. For example, ...’ I don’t understand this strategy, the fit of a set of parameters can be done simultaneously. In addition, what procedure was used? Some mean squared error minimization? How many data were used to fit each parameter? My worry is that too few data were used, like 1 empirical curve was used to fit 4 parameters (fig 2e,f). It’s not clear also why some of these parameters should be inferred per complex, instead of across complexes, for example the extension at zero force. This would allow the authors to use more curves to fit 1 parameter only, and in this way they could use some method like a leave-one-out cross validation.

I have then a few remarks in terms of clarity. In general, steps and findings are linked to published papers, apart from a few instances where I think a bit more explanation is needed, e.g.: 1. the use of

11.6nm as reasonable guess value for the N15 complex; 2. the expressions S8ab are not really justified. I would also move the summary table on the meaning of the different parameters to the main, it's quite important to follow the discussion.

RESPONSE TO REVIEWER COMMENTS

Reviewer #1:

I. Summary

Choi et al. lay out a mathematical approach to predict structural changes as they may occur within peptide presenting MHC class I and class molecules and their cognate $\alpha\beta$ TCRs upon complex formation within the confines of the immunological synapse. The work is inseparably tied the concept of catch-slip bonds, as they have been first described for stimulatory TCR-peptide/MHC (pMHC) interactions in 2014 by the corresponding author of this ms. Based on Kramer's kinetic rate theory the authors formulate a rationale, in which they assume a semi-rigid body nature of the TCR-pMHC complex (which was earlier predicted by the corresponding author via steered molecular dynamics (SMD) simulations) to calculate extensions of more flexible inter-domain linkers, dissociations of interdomain contacts, as well as inter-domain rotations under mechanical force. Such force-dependent structural changes are ultimately projected to create binding conditions under which catch-slip bonds take effect: force-directed tilting of the interaction surfaces between TCR- and pMHC. This behavior then serves as an explanation for what the authors term "catchiness". To validate their theoretical leads the authors correlate calculated catch-slip bond behavior with force-bond lifetime plots as they have resulted from carrying out two-dimensional BFP-based force-response assays. Given observed similarities between predicted and measured "catchiness", the authors present a mechanistic explanation of how the phenomenon of catch-slip bonds may allow T-cells to discriminate antigenic from endogenous pMHCs and also how TCRs are triggered upon extracellular ligand engagement to transduce signals across the plasma membrane.

We thank the reviewer for carefully reading our manuscript and for providing invaluable comments. We recognize the significant amount of time and effort that the reviewer must have put in to review our long, complex, and math-heavy manuscript for which we really appreciate, despite that our views differ from those of the reviewer's in several points.

II. Reviewer's concerns

Explaining how T-cell recognition proceeds on a molecular level is without much doubt a task of colossal proportions. As a reader and reviewer of the ms. I find the presented wealth of intriguing reasoning refreshing. Most importantly, this reasoning leads to testable hypotheses, which have the potential to transform our understanding of T cell recognition with palpable implications for clinical translation, if the validity of these hypotheses holds firm after rigorous wet lab testing.

We thank the reviewer for the positive comments regarding the significance of our work.

Unfortunately, the authors refrain from providing direct evidence for the proposed structural changes, which I consider the central element of their line of thinking. Instead, they base their data validation entirely on previously reported experiments or simulations, which do not qualify as direct proof. More specifically, the authors refer to (i) SMD simulations in combination with (ii) experimental findings derived either from thermo-fluctuation / force-lifetime assays (which are indirect in nature as they require a number of yet unverified assumptions, e.g. single molecule detection, lack of cellular changes after repetitive testing on the same T cell), or from (iii) the use of magnetic tweezer-based force cycles, which give rise to a noisy readout. The latter approach has so far lacked statistical corroboration, and solid conclusions on force-induced structural changes are hence not yet supported. Looking at all the evidence provided, I am not convinced of structural changes within the TCR-pMHC complex upon synaptic bond formation, as other explanations are still possible. My opinion would change if I were offered clear evidence, such as (single molecule) life cell microscopy reporting on the deformation of synaptic TCR-pMHC complexes (or TCR or pMHC). For example, experiments involving

intramolecular FRET within pMHC or the TCR could in real time and most directly confirm the kind of structural changes put forward in the ms.

We respectfully disagree with the Reviewer’s characterization of the wealth of published results from which we derived our model assumptions and with which we compared our model predictions. These results were collected from 9 papers by four laboratories – the Cheng Zhu lab of Georgia Tech [1-5] (plus new data added in this manuscript), the Wei Chen lab of Zhejiang University [6], the Brian Evavold lab of University of Utah [7-9], and the Matthew Lang lab of Vanderbilt University [10] – totaling 55 bond lifetime vs force datasets of 12 TCRs measured using not only biomembrane force probe (BFP, by the Zhu, Chen, and Evavold labs) but also optical tweezers (by the Lang lab). Consistent with these datasets that involve only TCR–pMHC interactions are 48 additional datasets that involve either pre-TCR–ligand interactions [11, 12] or TCR–pMHC–CD4/8 interactions [4, 5, 8, 9, 13], which also exhibit either catch-slip bonds or slip-only bonds, although we did not include them in the present paper because these bonds have different structures. Collectively, these represent the best available datasets measured using the best existing technologies in the field, which have directly demonstrated that TCR forms catch-slip bonds with strong ligands but slip-only bonds with weak ligands, therefore also showing that the TCR catch bonds are not artifacts because slip bonds were also observed using often the same TCR (or the same pMHC) and same experimental technique, with the only differences being altered ligands of lower potencies (or mutant TCRs of lower signaling capabilities).

Specifically, we disagree with the reviewer’s statement that the published measurements were “indirect in nature as they require a number of yet unverified assumptions, e.g. single molecule detection, lack of cellular changes after repetitive testing on the same T cell”. To the contrary, the assumption of single molecule detection has been rigorously and extensively validated in the Zhu lab publications on force-dependent bond lifetimes of TCR–pMHC interactions and of other receptor–ligand interactions. Importantly, in many of the publications, the Zhu lab employed no less than two experimental techniques, including not only atomic force microscopy (AFM) and BFP, but also laminar flow chamber to show catch bonds of selectins [14-16], integrins [17, 18], platelet glycoprotein Ib [19, 20] and Fc γ receptor IIA [21] in a technique-independent fashion. These techniques and results have been well accepted by the field of single-molecule biophysics.

The possibility that the bond lifetime may be affected by repetitive testing on the same T cell has also been tested in three ways [22]. In the first two tests, we asked whether bond lifetimes measured in the present contact cycle were influenced by the outcome of the immediate past contact cycle. Two outcomes were considered: 1) whether the immediate past contact cycle resulted in a binding event or not (**Fig. R1A**) and 2) whether the immediate past contact cycle resulted in a lifetime event or not (**Fig. R1B**). In the third test, we asked whether repeated contact cycles on the same T cell would result in changing bond lifetime over time or not (**Fig. R1C**). As exemplified by the data shown in **Fig. R1** [22], the answers to our questions are NO in all three cases.

Magnetic tweezers are generally accepted as the most stable and robust technique available nowadays for measuring conformational changes in DNA and proteins [23-25]. Structural destabilization and changes in the (pre)TCR–pMHC complexes have been observed by both the Lang lab using optical tweezers [10, 11, 26] and the Chen lab using magnetic tweezers and BFP [6].

Fig. R1. A, B. Lack of effect of the immediate past interaction on the current bond lifetime measurement. P14 TCR–gp33:H2-D^b bond lifetime measured on the surface of splenic CD8⁺ T cells after a binding (A) or lifetime (B) event (red) vs a non-binding (A) or non-lifetime (B) event (blue). C. Lack of effect of repetitive testing on the same T cell on the bond lifetime measurement. E8 TCR–TPI:HLA-DR1 bond lifetimes were measured from 5 T cells each repeatedly tested 500-1000 contact cycles, resulting in 598 bond lifetime measurements. For each cell the data were segregated into two groups, consisting of bond lifetimes measured from the first and second half of the contact cycles. For each group the bond lifetimes were pooled from all cells tested and plotted vs force, which are compared with each other and with all the data without segregation.

For these reasons, we respectfully reject the mischaracterizations of our published simulations and experiments, which we stand by.

Finally, while we appreciate the Reviewer's suggestion of the intramolecular FRET experiments, our manuscript describes a piece of modeling work. We understand and accept any requests for us to provide rationale/justifications for the model assumptions and comparison of the model predictions with existing experimental/simulation results, which we did extensively, thoroughly, and rigorously. However, we respectfully submit that it is not reasonable to demand us to perform new experiments in this modeling paper to further test the model assumptions and predictions, such as the intramolecular FRET experiments, because it is beyond the scope of the present theoretical work, which is already quite extensive in its scope.

Fig. R2. Model validation by mutagenesis. **A-C** Comparison of structures (**A, B**) or noncovalent contacts (**C**) of interactions of the H2-K^b $\alpha_1\alpha_2$ (blue) with mouse (**A, C**) and human (**B, C**) β_2m (purple for mouse and orange for human). The structures in **A** and **B** are depicted by ribbon diagram using snapshots from MD stimulations (based on 1G6R and 2BNR) with side-chains of the interacting residues shown to indicate their locations. Simulated time-courses of distances between the interacting H2-K^b α chain residues and β_2m residues are plotted in **C**, showing shorter distances with the human β_2m and longer distances with the mouse β_2m . **D** BFP measured bond lifetime vs force data (points, mean \pm sem) and model fits by $1/k(F)$ (curves) to of 2C TCR expressed on CD8⁺ naive T cells interacting with indicated R4:H2-K^b α 3A2 with either human or mouse β_2m . **E** Comparison of potencies to activate naive CD8⁺ 2C T cells by hybrid (left column) and WT (right column) R4:H2-K^b at 0.1 μ g/ml (upper) or 1 μ g/ml (lower) concentration for 72 hours. T cell activation was assayed by flow cytometric analysis of upregulation of surface markers CD69 (y-axis) and CD25 (x-axis) using PE-conjugated anti-CD69 and PE-cy7-conjugated anti-CD25 antibodies. **F** Structure of 2B4 TCR $\alpha\beta$ showing the locations of residues N222-P223, G182-K183, and N225-H226 on C β domain with CD3 complex (6JXR). **G-H** Comparison of interactions of P223 (1st row), K183 (2nd row), and N225 (3rd row) with the corresponding CD3 residues in the absence (**G**) and presence (**H**) of force. **I** Simulated time-courses of distances between C β P223 and CD3 ϵ L90 (1st row), C β K183 and CD3 ϵ L90 (2nd row), as well as C β N225 and CD3 γ E38 (3rd row) in the absence (gray) and presence (colored) of force. **J** Relative fold-change of effector function of 2B4 hybridoma using the area under the dose response curve (AUC) of the WT and indicated mutant 2B4 hybridoma IL-2 production. **K** BFP measured bond lifetime vs force data (points, mean \pm sem) fitted by $1/k(F)$ predicted by the TCR-pMHC-II model (solid curves) of WT and indicated mutant 2B4 TCRs on hybridomas interacting with K5:I-E^k.

That said, we have added in the revised manuscript results from two sets of new mutagenesis studies to provide further validation of both our class I and II models, respectively, using mutations located away from the TCR and pMHC binding interface but capable of impacting their respective conformational changes under force, which were analyzed by MD simulations, bond lifetime measurements, and functional assays. These new results are presented in Fig. 3b part of 1st panel, Fig. 5d lower panel, and Fig. 7, which are collectively presented here as Fig. R2.

The first set of studies compared the WT and a hybrid H2-K^b that swaps the mouse β_2m with the human β_2m because the latter binds the mouse class I heavy chain with a higher affinity and better support peptide binding than the former [27]. Since it is easier to make soluble hybrid than complete mouse H2-K^b protein, many of our previous studies used the hybrid H2-K^b (Supplementary Table 3). Surprisingly, T cells kill less efficiently target cells expressing the hybrid H2-K^b than the WT molecule [28]. Our previous study using double-cysteine mutations to lock the $\alpha_1\alpha_2$ - β_2m connection by disulfate bond suppressed both pMHC conformational changes and its catch bond with TCR concurrently [6] (Supplementary Fig. 1b, compared the R4 curves in panels 2 and 3). Using SMD simulations, we observed force-induced dissociation of the $\alpha_1\alpha_2$ - β_2m interdomain bond (Supplementary Movie 1). We compared MD simulated interactions of H2-K^b α chain with mouse β_2m (using the crystal structure 1G6R) and human β_2m (using a model built based on 1G6R and 2BNR), finding that Arg14, Glu232, and Gly237 of the H2-K^b α chain respectively interacted with three residues – Asp34, Lys6, and Tyr67 – of the human β_2m but not the corresponding residues of the mouse β_2m (**Fig. R2A-C**). This indicates that the hybrid H2-K^b has a more stable structure and hence less able to respond to force induction of conformational change than the WT molecule, predicting a less pronounced TCR catch bond with the same peptide presented by the hybrid than the WT H2-K^b. Remarkably, the newly measured force-dependent bond lifetime indeed showed a much more pronounced catch bond of the 2C TCR with R4 peptide bound to WT than hybrid H2-K^b (**Fig. R2D**), supporting the prediction of our class I model. Consistent with previous report [28], functional assay also showed that the WT H2-K^b was more able to activate T cells than hybrid H2-K^b (**Fig. R2E**), further validating the class I model.

The second set of studies examined a hybrid TCR with the mouse 2B4 $V\alpha\beta$ fused with the human LC13 $C\alpha\beta$ and 4 double mutations on the $C\beta$ domain, which have been indicated by our previous NMR and chemical shift experiments [29] and by recently published cryoEM structures [30, 31] to impact its interactions with human CD3 (**Fig. R2F**). We performed MD simulations to examine the $C\beta$ -CD3 *cis*-interactions in the absence (**Fig. R2G**) and presence (**Fig. R2H**) of a force to mimic pulling on the $V\alpha\beta$ by the engaged K5:I-E^k (Supplementary Movie 2-4). We found that $C\beta$ Pro223 is force-stabilizing (**Fig. R2I, top**) whereas $C\beta$ Lys183 and Asn225 are force-destabilizing (**Fig. R2I, middle and bottom**). These results suggest that the double mutant N222A/P223A (NP) may result in less stable, whereas G182A/K183A (GK) and N225A/H226A (NH) may result in more stable, $C\beta$ -CD3 *cis*-interactions under force. Consequently, the NP mutant may put more restriction, whereas the GK and NH mutants may put less restrictions, respectively, on force-induced conformational changes in the TCR $\alpha\beta$ than the WT molecule. Interestingly, NP was identified as a gain-of-function mutation whereas GK, NH and HN (plus another double mutant H221A/N222A, or HN) were identified as loss-of-function mutations by functional assays (**Fig. R2J**) [29]. Supporting the prediction of our class II model, force-dependent bond lifetime measurements by BFP indeed showed a more pronounced catch-slip bond of the NP mutant, less pronounced catch-slip bonds of the GK and HN mutants, and a slip-only bond for the NH mutant, compared to the WT 2B4 TCR interaction with K5:I-E^k (**Fig. R2K**). Remarkably, the bond profile metrics (Fig. 5e and Supplementary Fig. 8, *blue*) and best-fit model parameters (Fig. 5f *blue* and 5g *green*) were found to correlate with T cell function, further validating the class II model.

Yet this direct experimental evidence is so urgently needed to resolve many of the controversies in the field, which is - unlike what the authors wish to portray - divided into a “catch-slip bond-” and

“slip-bond-camp” and a large group of scientists who do not know whom to believe. What adds to my concerns is that fact that the authors willfully ignore findings by others that challenge the core of the catch-slip bond hypothesis or the relevance of catch-slip bonds for TCR-proximal signaling. To name 2 of many examples, there is no mentioning of the (i) bead-based experiments done in laminar flow by Philippe Robert and colleagues, who failed altogether to directly observe any catch-bond behavior of 5 well-studied TCR-pMHC pairs (PMID: 31315981). (ii) The authors do not cite the force measurements by Schutz and his team on single TCRs engaging pMHCs or anti-TCR scFVs (PMID: 33947864). These single molecule measurements were conducted within the confines of the area of contact between T cells and stimulatory planar glass-supported lipid bilayers, and measured forces amounted to less than 2pN for TCRs bound to bilayer-attached anti-TCR scFV, i.e. 10-times lower forces than the forces that catch bonds are reported by Zhu and colleagues to require in order to exert their effect on TCR-pMHC lifetimes. Of note, the force loading rate measured by the Schutz group amounted to about 1.5 pN/s which renders it highly unlikely to arrive at 10-15 pN forces with the rather short pMHC-TCR lifetimes measured by means of a BFP (PMID: 24725404), single dye tracing (PMID: 23840928) or FRET-based recordings (PMID: 20164930).

We acknowledge the existence of controversial views in the field regarding whether TCRs would form catch bonds with some pMHCs and whether T cells exert endogenous forces on TCRs, as reflected in the two papers cited by the reviewer. Specifically, we are aware of the paper by Philippe Robert and colleagues [32] but considered their results unreliable because of several technical limitations and/or misinterpretation of data: i) limited data: only a single TCR (1G4) was studied, and more critically, only two data points (at 6 and 10 pN) were measured at low forces for all five ligands tested; ii) incorrect readouts measured by the flow chamber technique; and iii) misinterpreting the inability to find evidence of existence as evidence of nonexistence.

By comparison, the Chen lab has also studied the same 1G4 TCR and found it formed catch-slip bonds with five ligands [6]. Importantly, all these catch bonds occurred in forces ≤ 10 pN and the authors measured 3-4 points in the low force regime by the BFP to ensure the catch trends are reliable [6].

FIGURE REDACTED

As mentioned earlier, the Zhu lab has published extensively studies combining flow chamber with AFM and BFP on selectins [14-16, 33] and platelet GPIb α [19, 20, 34] that respectively mediate neutrophils and platelets tethering to and rolling on endothelial cells under flow. In contrast to these relatively strong interactions, binding of TCR on flowing T cells to pMHC immobilized on the chamber floor is extremely weak and brief, manifesting as only slight reductions of the velocity of the cells moving through the flow chamber in a manner that is correlated to the peptide potency [FIGURE REDACTED] with only a tiny fraction of cells being arrested at the pMHC surface [FIGURE REDACTED]. These transient interactions are biologically relevant as they induce intracellular calcium after the cells pass through the pMHC surface [FIGURE REDACTED]. Importantly, the level of velocity reduction correlates with the level of calcium signal [FIGURE REDACTED]. Unlike catch bonds of selectins and GPIb α that reduce the magnitude of the velocity of neutrophils and platelets rolling on ligand-coated surface as shear increases [20, 35], the TCR-pMHC catch bonds reduce the increase in the velocity V_{pMHC} (relative to the BSA control V_{BSA}) with increasing shear of T cells moving through pMHC surface [FIGURE REDACTED]. To visualize the TCR-pMHC catch bond in flow chamber, we multiply the specific reciprocal velocity $(1/V_{\text{pMHC}} - 1/V_{\text{BSA}})$ by a characteristic length L to calculate the time required for the cell to travel through this distance, $t_c = L(1/V_{\text{pMHC}} - 1/V_{\text{BSA}})$. This t_c should be proportional to the TCR-pMHC bond lifetime because the longer the bond lifetime, the slower the velocity. We found that this surrogate bond time for the OT-1 TCR-OVA:H2-K b interaction increases with force on a tether bond until 12.5 pN, and decreases with further increase in force thereafter [FIGURE REDACTED], whereas the t_c vs force curve of the OT-1 TCR-G4:H2-K b interaction shows a slip bond characteristic [FIGURE REDACTED]. Both datasets are consistent with what we previously measured by BFP [1]. Note that we used a 50 μl syringe to generate 0.01 $\mu\text{l}/\text{min}$ flowrate in a microfluidic channel, yielding

3 data points in the catch bond regime to ensure reliable observation of the catch bond trend [FIGURE REDACTED].

Apparently, Robert and colleagues measured the duration of arrest from the fraction of beads that adhered to the chamber floor by the TCR–pMHC interactions as their experimental readout whereas we focused on the fraction of moving cells [FIGURE REDACTED]. In our opinion, the data by Robert and colleagues are of much lower quantity and quality than the data we used to test our models in our manuscript. We would have doubted their data less and discussed the view of the paper cited by the reviewer [32], if they had included a positive control to show that the absence of catch bonds from their experiments was not due to technological limitations, which prevented them from detecting catch bonds. Unfortunately, they did not do that. We note that in a recent biorxiv preprint from Robert and colleagues, the authors observed a catch bond between the A6 TCR and 7Q pMHC with a much broader catch bond force regime using their flow chamber technique [36].

We have also noted the discrepant results (2 pN vs. 12-19 pN) of the work cited by the reviewer [37] and the multiple studies by the Salaita lab and us [4, 38-41], measuring the endogenous forces exerted by T cells on TCR. The Schutz group used a spider silk peptide-based tension probe [37], whereas the Salaita lab and us used DNA hairpin-based tension probe [42]. We suspect that the discrepancies may be due to the different force probes used, which has different force responses.

Both probes behave as nonlinear springs but the DNA spring shows much higher level of nonlinearity than the spider silk peptide. **Fig. R4** shows the force-extension curves calculated using an extensible worm-like chain model [43-45], predicting that it would require 4.7 nm of TCR retraction away from the force probe functionalized glass surface to generate 12 pN force in the spider silk peptide-tension probe, but 9.4 and 9.8 nm, respectively, of TCR movements to activate the 4.7 and 12 pN DNA hairpin-based tension probes, respectively. Consider the scenario where cells actively apply forces on TCR, which depends largely on myosin II and cytoskeletal components [3]. Assuming that a myosin II molecule could move ~5 nm per step [46] and generate ~3.5 pN force [47], it would require ~3 myosin II molecules to move a single step synchronously to generate >10 pN force in the spider silk peptide-based force probe. By comparison, it follows from the DNA force-extension curves that the first 9 nm of TCR retraction would encounter ≤ 3 pN resistance, which can be achieved by a single myosin II motor that moves 3 steps, allowing it to recruit 1 or 2 more myosin II molecules to join force to overcome the remaining resistance to open a DNA hairpin of 4.7- or 12-pN force thresholds, respectively. This line of reasoning suggests that the T cells may react differently to the two types of force probes to generate difference forces.

Fig. R4. Force vs. extension curves of spider silk peptide (red) and DNA hairpins that unzips at 4.7 pN (cyan) and 12 pN (blue).

Perhaps more importantly, the spider silk peptide-based force probes have an analog response where the peptide extends instantaneously with, and proportional to, force (with the spring constant as the proportionality constant). This means that if cytoskeleton and motor-powered forces are applied as periodical brief impulses (which is likely considering how the cytoskeleton and involved motors work), the FRET signal might have been missed or averaged out over the exposure time. In contrast, DNA-based tension probes work in a digital fashion, requiring an above threshold force to unfold the hairpin. Even as an impulse, it would break the hairpin and transform it from the signal-off state to the signal-on state. Refolding of the hairpin is relatively slow and this property has been taken advantage to use complementary DNA in solution to lock the unfolded DNA hairpin in the signal-on state [48]. Therefore, the DNA-based tension probes can capture such events where large but brief forces are applied, which could have been averaged out or even missed by the spider silk peptide-based FRET probes. We note

that similar discrepancies (a few pN vs. tens of pN) have been reported between the Dunn lab [49] who used the spider silk peptide-based tension probe and three other labs [41, 50-57] and us [42] who used DNA-based tension probes. Importantly, TCR forces have been measured using traction force microscopy [58, 59] and micropattern array detectors [60, 61]. Also, integrin forces have been extensively studied using these two assays [62, 63]. We further note that traction force microscopy and micropattern array detectors lack single-molecule sensitivity and hence could only measure sufficiently large forces, which are consistent with the results obtained using DNA-based force probes but not spider silk peptide-based force probes. We therefore respectfully submit that our results agree with a large number of publications and the paper cited by the reviewer [37] is at odds with the general consensus of the field.

Despite our above opinions, we agree that we should cite and discuss the two papers mentioned by the reviewer [32, 37], which we have done in the Discussion of the revised manuscript.

While these two studies do not necessarily rule out the existence of TCR-pMHC catch bonds or their relevance for antigen recognition (also in view of much higher force values measured by Salaita's team, PMID: 27140637), they certainly need to be reckoned with within the context of his ms.. In essence, such findings demand that TCR-pMHC catch bonds as well as the structural changes that may cause them to take effect (as elaborated in this ms.) need to be demonstrated most directly (and not indirectly which would leave room for alternative explanations) as a means to establish them beyond doubt as a principle underlying T cell antigen recognition and ligand discrimination.

We appreciate the reviewer's acknowledgement that the two studies by Robert and colleagues [32] and the Schutz group [37] "do not necessarily rule out the existence of TCR-pMHC catch bonds or their relevance for antigen recognition". We also agree that TCR-pMHC catch bonds need to be further studied "to establish them beyond doubt as a principle underlying T cell antigen recognition and ligand discrimination". However, we respectfully submit that these experimental studies should be done in the future, not in the present manuscript, which reports modeling work and hence should be judged by the models' own merit.

In the absence of such demonstration and a more balanced description of the current state of findings the ms. runs the risk of being perceived as "prediction in hindsight", in particular since its findings are predominantly based on data gathered by the lab of the corresponding author of this ms.. Also, the ms. would certainly gain in credibility by establishing data transparency. This may include providing the means to reproduce the mathematical proceedings and making unprocessed primary data (e.g. BFP-based experiments, magnetic tweezer experiments) available for download.

The point of a more balanced description is well taken, which has been added to the revised manuscript. The point of data transparency is also well taken. We have provided extensive Supplementary Methods that thoroughly describe the derivation of all equations, added detailed data fitting procedures, and tables summarizing all data used to fit the models and the fitting parameters. The individual data points for all bond lifetime vs force datasets will be deposited in Github (<https://github.com/Chengzhulab/NCOMMS-22-20167>).

Taken together these considerations (lack of direct and clear evidence for structural changes, overstatements regarding the implications of the findings/results for mechanisms underlying TCR-based pMHC discrimination and signaling, highly unbalanced description of the state of research) prevent me from recommending publication of this study in Nature Communications. Given the emphasis on mathematical modeling, and the effort it takes the average life scientist to understand the content, the ms. may be more suitable for a more specialized journal targeting physicists or life scientists with an extensive physics background.

Again, we respectfully and strongly disagree with the reviewer's characterization of the published data by us and others supporting the assumptions and predictions of our models. Our rationale has been presented above, including detailed analyses of the two papers cited by the reviewer. We agree that we should mention these contradictions to the majority view for a more balanced presentation, but they should not be the reason for us not to pursue the modeling work presented here.

III. Specific points

1. The ms. is well structured. Overall the figures are clear. See last point (Suggestions).

We thank the reviewer for the positive comment.

2. Experimental details (temperatures, constructs) should be indicated more clearly for all listed experiments. I found myself too often looking these data up in the literature.

The data were generated by the original experimental papers and were used in the present manuscript for comparison to our model predictions. Nevertheless, we have added the construct details to the legends of figures and tables wherever appropriate. All experiments were done at room temperature.

3. Confidence intervals or error bars are missing in the following figures: Fig. 1c, e, g, Fig. 2c, d, f, e, Fig. 3b, Fig. 4c.

Fig. 1 (new Fig. 2) panes c, e, and g are model-prediction, which are not supposed to have error bars. The R^2 values for Fig. 2 (now Fig. 3) panels c and d have already been provided. The catch-bond intensity values in Fig. 2e and Fig. 3b are defined from each force-lifetime curve as a single value and hence no error bar. The best-fit n^* values in Fig. 2f, middle panel are integers selected based on the method shown in Supplementary Information (Supplementary Table 2 and the related text), which has no error bars.

4. How do the authors arrive at the demarcation separating Catch-only, Catch to slip and slip-only in Fig. 1c lower panel?

We firstly changed one parameter at a time (either the tilting angle θ or the number of unfolded amino-acid n^*) while keeping other parameters constant to examine how these parameters control our model behaviors individually (Fig. 2c). This step informed us where in the parameter space the TCR-pMHC complex has catch-to-slip dynamics. If there is a biphasic transition in extension-change vs force plot, catch behavior would be observed. For example, in the lower panel of Fig. 2c, solid line in Set 1 indicates 'catch-to-slip' while dotted line indicates 'catch-only'. Moreover, dotted line and solid line in Set 2 show 'slip-only' and 'catch-to-slip', respectively. In the "phase diagram" of the parameter space, the number of amino-acid is integer, thus the demarcation separating catch-only, catch to slip and slip-only displays a stepwise/discrete pattern (Fig. 2d lower panel).

5. Some of the wording in the Supplementary Information section needs editing.

We have edited in the Supplementary Information in the revised manuscript to increase readability.

6. Specific examples of wording which I find problematic

- Title:

"Catch bond models explain how force amplifies TCR signaling and antigen discrimination"
The wording is too strong, as the work provides testable yet untested hypotheses. Changing the title to "Catch bond models may explain how force amplifies antigen discrimination" or similar would be more adequate.

We have revised the title to be: "Catch bond models may explain how force amplifies TCR signaling and antigen discrimination" based on the reviewer's suggestion.

- Abstract:

"Central to T cell biology, the T cell receptor (TCR) integrates forces in its triggering process upon interaction with peptide-major histocompatibility complex (pMHC)"

The wording is too strong and projects an unbalanced view of the current state of data.

As discussed in detail in the preceding responses to the reviewer's main concerns, it is a majority view of the field that force plays a role in TCR triggering process. Note that we carefully chose the word "integrate" instead of "require" and the phrase of "triggering process" instead of just the word "triggering" to indicate that it is not necessary the initiation step, but a time process during which force can play a role.

"Phenotypically, forces elicit TCR catch-slip bonds with strong pMHCs but slip-only bonds with weak pMHCs While such correlation is generally observed..."

The wording is too strong / unbalanced and does not capture the current state of ground truth finding.

We stand by this statement based on reasons discussed in detail in the preceding responses to the reviewer's main concerns.

"The extensive comparisons between theory and experiment allowed us to validate the models and identify specific conformational changes that control bond profiles, thereby providing structural insights into the inner workings of the TCR mechanosensing machinery and explaining why and how force amplifies TCR signaling and antigen discrimination."

The wording is too strong and overstates the results.

The comparisons between theory and experiment as well as model validations performed in this paper are by far the most extensive, which both Reviewers #2 and #3 agreed as stated in their positive comments (see below). We did identify specific conformational changes that control bond profiles in the models. These results therefore provide structural insights into the inner workings of the TCR mechanosensing machinery and explained why and how force amplifies TCR signaling and antigen discrimination, again, in the models. So what we said were not overstating and not too strong, but are accurate descriptions of what have been done in the paper.

Note that we did not say the comparisons and validations are complete and finished, hence need no more in the future. We also did not say that these hypothetical conformational changes have been fully observed by extensive experiments. We did not say that the structural insights we obtain must be truth.

Nevertheless, to avoid any confusion, we have revised the statement as follows:

"The extensive comparisons between theory and experiment provided strong validation of the models and testable hypotheses regarding specific conformational changes that control bond profiles, thereby suggesting structural mechanisms for the inner workings of the TCR mechanosensing machinery and plausible explanations of why and how force amplifies TCR signaling and antigen discrimination."

- Introduction:

"Mechanical forces applied to TCR via engaged pMHC substantially increase antigen sensitivity and amplify antigen discrimination"

A large part of the community will not agree with this strong statement.

Evidence for this statement has been provided by at least 9 papers [1, 3, 4, 6, 7, 10, 13, 26, 64]. We are not aware of a single paper that offers evidence that counters this statement. The reviewer cited a paper by Robert and colleagues that challenges the existence of TCR –pMHC catch-slip bond [32]. But it has shown by the Schutz group that force improves ligand discrimination by the TCR even in the case in which the TCR forms slip-only bonds with pMHCs [64]. The reviewer cited another paper by the Schutz group suggesting the lack of T cell forces on TCR in the immunological synapse [37]. However, this does not negate the statement because, if mechanical forces are applied to the TCR via engaged pMHC, this would substantially increase antigen sensitivity and amplify antigen discrimination.

“As a fundamental force-elicited characteristic, strong cognate pMHCs form catch-slip bonds with TCR where bond lifetimes increase with force until reaching a peak, and decrease as force increases further, whereas weak agonist and antagonist pMHCs form slip-only bonds with TCR where bond lifetimes decrease monotonically with increasing force.”

There is plenty of evidence challenging these bold statements.

We stand by this statement for reasons discussed in detail in the previous response to the reviewer’s main concerns.

7. Suggestion

If targeting a broad life science readership is the aim, it may be helpful to provide a graphic explanation of the mathematical reasoning behind model development.

This is an excellent suggestion! We sincerely thank the reviewer for this and have added as Fig. 1 a graphic explanation of the mathematical reasoning behind model development.

Reviewer #2:

In this manuscript, Choi and colleagues develop models for TCR–pMHC complex interactions, and in particular description of catch-slip bond and slip bond dynamics. The models are robustly tested across a number of datasets and experimental data is well used to develop key aspects of the models. The models do provide structural and mechanical insights into TCR interaction behaviors.

Overall, I commend the authors on the amount of model development, rigor (and rigorous testing) and large-scale testing with multiple unique datasets.

We thank the reviewer for the positive comments and for his/her carefully reading this complex and math-intense manuscript.

However, this reviewer is struggling to figure out how impactful and unique the new models and findings really are. Without direct comparison to the “next best” models it is extremely difficult to understand how much of an advance these models really are in terms of understanding unique aspects of TCR–pMHC mechanobiology.

The reviewer’s point is well taken. In response, we have employed another model – an existing two-pathway catch-bond model [65] (Fig. 1a, Eq. 4) – that has been applied to the N15 TCR–pMHC systems by others [10] to analyze the same experimental data. This additional model analysis and the results have been incorporated into the revised manuscript. In brief, the alternative model also fit all the

experimental force-lifetime datasets with statistically indistinguishable goodness-of-fit compared to our models (Supplementary Fig. 1 and Fig. 5d). However, neither can this 'next best model' distinguish class I and class II MHC systems nor do its best-fit parameters correlate with the potency of the TCR or pMHC for their ability to trigger T cell activation (Supplementary Fig. 6 and Supplementary Table 4), which our models can do in both counts. These new results strengthen our paper for which we thank the reviewer.

That said, there are cases where it appears clear that the models are an advance since the assumption is that published models would not be able to describe the data – however this is not well described by the authors in most cases. Likewise, the biological implications of the model findings are not robustly presented. For a general audience such as Nature Communications it would strengthen the work to make it very clear where the advances are and what new information is being gained. Along those lines, while the models are very elegant and insightful, the finding don't bring the title to realization and explain how force amplifies TCR signaling and antigen discrimination. As such, this leaves a question for the Editor as to whether or not Nature Communication is the appropriate choice versus a more specialized journal.

The reviewer's point is well taken. In the revised manuscript we have better articulated why and how our models are a significant advance. This is done by benchmarking with the "next best model. We have also better presented the biological implications of the model findings.

This reviewer does not have any technical concerns from the model formulation, testing, and application. At times the model development is dense to work through and not particular accessible to a general audience, but a deep dive into the model reveals it rigor.

We thank the reviewer for the positive comments on the rigor of our work.

A few more minor comments:

1) The justification for the model needs on page 3 is not particularly compelling. It is also not clear why the authors consider catch-slip bond counterintuitive. When considering many of the key behaviors addressed in the manuscript they are quite intuitive.

People in the field consider slip bonds intuitive as force is expected to disrupt structure, hence accelerating bond dissociation and shortening bond lifetime. Catch bonds behave in an opposite way, decelerating bond dissociation and prolonging bond lifetime. Hence people consider catch bonds counterintuitive.

2) as discussed above the implications of the model findings are not well developed.

This has been addressed in the revised manuscript.

3) it is not clear how the authors arrive at 48 datasets. Perhaps a table with all the datasets and what is being tested would be beneficial.

This is a good suggestion. We have added explicit descriptions of these as TCR-pMHC bond lifetime vs force datasets published to date in 9 papers by four laboratories measured using two different techniques, including 55 datasets of 12 TCRs and their mutants interacting with corresponding panels of both classes of pMHCs without coreceptor engagement. These TCR-pMHC bond bonds have been summarized in Supplementary Tables 3 (class I) and 6 (class II) along with their best-fit model parameters.

4) (very minor), there are a number of typos and at times model terms are not defined on first mention.

We thank the reviewer for pointing this out. We have fixed the typos.

Reviewer #3:

The authors propose a model of the catch and slip-bonds based on a one-dimensional (1D), single-well energy landscape, whose mathematical behaviour was developed in a previous publication (Guo et al.). In this article, the authors apply this model to describe TCR-pMHC complexes, exploit available data on the bond lifetime vs force relationship to assess how well such a model can explain it and can predict peptide potency, i.e. the strength of signalling upon TCR binding to a peptide.

In my opinion the work is valuable, the authors have for sure carried out an extensive and thorough analysis, however I am not convinced about how solid the main claim is, i.e. the claim that the model explains mechanistically and quantitatively catch-slip bonds in TCR-pMHC complexes. This claim is supported by the fit to data and by the correlation of 4 parameters (and not k_0) with peptide potency. My main concern (and I think the authors should try to explain this point very clearly) is the rationale by which one should expect a correlation of 4 parameters (and not k_0) with peptide potency and why instead the absence of this correlation can be taken as a way to select against a certain model. This correlation to potency (fig. 2f) should be represented by some scatter plot, giving a quantitation of correlation coefficient and its p-value. In addition, the fitting of models parameters to data is very unclear to me, it seems to me that the authors simply say 'The curve-fitting strategies involve varying one parameter while keeping others constant. For example, ...' I don't understand this strategy, the fit of a set of parameters can be done simultaneously. In addition, what procedure was used? Some mean squared error minimization? How many data were used to fit each parameter? My worry is that too few data were used, like 1 empirical curve was used to fit 4 parameters (fig 2e,f). It's not clear also why some of these parameters should be inferred per complex, instead of across complexes, for example the extension at zero force. This would allow the authors to use more curves to fit 1 parameter only, and in this way they could use some method like a leave-one-out cross validation.

We thank the reviewer for carefully reading our manuscript and providing valuable comments.

Regarding the concern on the main claim- In response to a comment of Reviewer #2, we used existing generic two-pathway model to analyze the same datasets. We showed that, whereas the generic model fits the force-dependent bond lifetime data equally well, neither can it distinguish class I and class II MHC systems nor do its best-fit parameters correlate with the potency of the TCR or pMHC for their ability to trigger T cell activation, which our models can do in both counts. This new finding further supports our main claim. Nevertheless, we have softened the statement to be "plausible explanations".

Regarding why k_0 was not used as a fitting parameter- It has been noted by us and others that Excedifferent TCRs have very different peak lifetimes, $t_{\text{peak}} = t_0 + \Delta t$, $t_0 = 1/k_0$, $\Delta t = 0$ for slip-only bond where t_{peak} occurs at $F = 0$, and $\Delta t > 0$ for catch-slip bond where t_{peak} occurs at $F = F_{\text{opt}} > 0$ (see Fig. 3a). While comparisons of bond lifetimes at zero force (t_0) and peak bond lifetimes (t_{peak}) at optimal force (F_{opt}) of the same TCR interacting with different pMHC ligands have generated interesting and useful information, comparing distinct TCRs with unrelated specificities have not. Examples for this point have been discussed in our manuscript. Specifically, a recent study reported that a pMHC ligand, NP₃₆₆:H-2D^{bD227K}, forms catch-slip bonds with TCRs B13.C1 and B17.C1 and induces T cell signaling, whereas the same pMHC forms slip-only bonds with TCRs B17.R1 and B17.R2 and does not induces T cell signaling [8]. However, the B17.C1 TCR-NP₃₆₆:H-2D^{bD227K} bond was shorter-lived than the B17.R2 TCR-NP₃₆₆:H-2D^{bD227K} bond across the entire force range tested. Even at 9.4 pN, which is F_{opt} for the

former with a $t_{\text{peak}} = 0.61$ s, the latter lived 2.48 s on average, and the longest lifetime of the latter is $t_0 = 2.83$ s that occurs at zero force [8]. As discussed in the manuscript, non-dimensionalized relative parameters work much better for comparison of bond profiles across different TCRs, and k_0 serves as a reference time scale for non-dimensionalization, which is why it is not a fitting parameter *per se*.

Regarding fitting data with four model parameters- It is true that we used models of four parameters to fit each bond lifetime vs force datasets. We should note that all published catch bond models have even more parameters (no less than 4 and as many as 10). In the section of 'Model Prediction', we changed one parameter while keeping the others constant to investigate the model behavior. In curve fitting, all four parameters were changed simultaneously to search for the minimum of the chi squared error. Of the 55 datasets analyzed, only one has 4 data points and this dataset shows slip bond; as such, is governed by two fitting parameters because the other two parameters are nearly zero. All other datasets have 6-10 data points; therefore, over-fitting is not a problem. We have added this discussion to Supplementary Model Derivation, A.4. Model applications, curve-fitting strategies, and biological relevance, of the revised manuscript.

Regarding the correlation to potency- Each bar in Fig. 2f (new Fig. 3e) represents a single value from fitting the data. Once the fitting parameters are determined by an appropriate n^* (total number of unfolded amino acids) as well as minimum RSS, the error bars of the best-fit parameters (or their ranges) were calculated from the standard errors of mean of bond lifetimes and the residual matrices. We have revised Supplementary Fig. 10 to show individual values of RSS and chi-square test for each model.

Regarding the reviewer's comment 'This correlation to potency (fig. 2f) should be represented by some scatter plot, giving a quantitation of correlation coefficient and its p-value', we also plotted the potency of ligand vs the best fitting parameters in OT1 TCR case (in new Fig.3f) showing quantitatively positive correlation. [Since we could not obtain functional data for all TCRs (but their potencies have been reported qualitatively by previous studies), we plotted correlation of the best-fit parameter vs the intensity of catch bond instead of functional data.]

Regarding the fitting strategies- We apologize for the lack of details in this section, which have been described in the Supplementary Model Development, A.4. We have indicated this in the main text of the revised manuscript.

Regarding fitting the model for data per complex, instead of across complexes- The extensions at zero force were measured from crystal structures of the TCR-pMHC complexes where they are available, and the value varies from structure to structure. Different TCR-pMHC complexes also have different biophysical characteristics (bond profiles) and induce different T cell functionalities. Our goal is to investigate whether, how, and why the bond profiles and functionalities can be captured by our models in terms of structural and biophysical parameters. Hence these parameters should be evaluated for each complex and then be examined across complexes.

I have then a few remarks in terms of clarity. In general, steps and findings are linked to published papers, apart from a few instances where I think a bit more explanation is needed, e.g.: 1. the use of 11.6nm as reasonable guess value for the N15 complex; 2. the expressions S8ab are not really justified. I would also move the summary table on the meaning of the different parameters to the main, it's quite important to follow the discussion.

We thank the reviewer for these suggestions and we revised the manuscript accordingly.

References

1. Liu, B., W. Chen, B.D. Evavold, and C. Zhu, Accumulation of dynamic catch bonds between TCR and agonist peptide-MHC triggers T cell signaling. *Cell*, 2014. **157**(2):357-68.
2. Liu, B., W. Chen, K. Natarajan, Z. Li, D.H. Margulies, and C. Zhu, The cellular environment regulates in situ kinetics of T-cell receptor interaction with peptide major histocompatibility complex. *Eur J Immunol*, 2015. **45**(7):2099-110.
3. Hong, J., S.P. Persaud, S. Horvath, P.M. Allen, B.D. Evavold, and C. Zhu, Force-regulated In situ TCR-peptide-bound MHC class II kinetics determine functions of CD4+ T cells. *J Immunol*, 2015. **195**(8):3557-64.
4. Hong, J., C. Ge, P. Jothikumar, Z. Yuan, B. Liu, K. Bai, K. Li, W. Rittase, M. Shinzawa, Y. Zhang, A. Palin, P. Love, X. Yu, K. Salaita, B.D. Evavold, A. Singer, and C. Zhu, A TCR mechanotransduction signaling loop induces negative selection in the thymus. *Nat Immunol*, 2018. **19**(12):1379-1390.
5. Rushdi, M.N., V. Pan, K. Li, S. Travaglini, H.-K. Choi, J. Hong, F. Griffiths, P. Agnihotri, R.A. Mariuzza, Y. Ke, and C. Zhu, Cooperative binding of T cell receptor and CD4 to peptide-MHC enhances antigen sensitivity. *Nat Commun*, 2022. **13**:7055.
6. Wu, P., T. Zhang, B. Liu, P. Fei, L. Cui, R. Qin, H. Zhu, D. Yao, R.J. Martinez, W. Hu, C. An, Y. Zhang, J. Liu, J. Shi, J. Fan, W. Yin, J. Sun, C. Zhou, X. Zeng, C. Xu, J. Wang, B.D. Evavold, C. Zhu, W. Chen, and J. Lou, Mechano-regulation of Peptide-MHC Class I Conformations Determines TCR Antigen Recognition. *Mol Cell*, 2019. **73**(5):1015-1027 e7.
7. Sibener, L.V., R.A. Fernandes, E.M. Kolawole, C.B. Carbone, F. Liu, D. McAfee, M.E. Birnbaum, X. Yang, L.F. Su, W. Yu, S. Dong, M.H. Gee, K.M. Jude, M.M. Davis, J.T. Groves, W.A. Goddard, 3rd, J.R. Heath, B.D. Evavold, R.D. Vale, and K.C. Garcia, Isolation of a Structural Mechanism for Uncoupling T Cell Receptor Signaling from Peptide-MHC Binding. *Cell*, 2018. **174**(3):672-687 e27.
8. Zareie, P., C. Szeto, C. Farenc, S.D. Gunasinghe, E.M. Kolawole, A. Nguyen, C. Blyth, X.Y.X. Sng, J. Li, C.M. Jones, A.J. Fulcher, J.R. Jacobs, Q. Wei, L. Wojciech, J. Petersen, N.R.J. Gascoigne, B.D. Evavold, K. Gaus, S. Gras, J. Rossjohn, and N.L. La Gruta, Canonical T cell receptor docking on peptide-MHC is essential for T cell signaling. *Science*, 2021. **372**(6546).
9. Kolawole, E.M., R. Andargachew, B. Liu, J.R. Jacobs, and B.D. Evavold, 2D Kinetic Analysis of TCR and CD8 Coreceptor for LCMV GP33 Epitopes. *Front Immunol*, 2018. **9**:2348.
10. Das, D.K., Y. Feng, R.J. Mallis, X. Li, D.B. Keskin, R.E. Hussey, S.K. Brady, J.H. Wang, G. Wagner, E.L. Reinherz, and M.J. Lang, Force-dependent transition in the T-cell receptor beta-subunit allosterically regulates peptide discrimination and pMHC bond lifetime. *Proc Natl Acad Sci U S A*, 2015. **112**(5):1517-22.
11. Das, D.K., R.J. Mallis, J.S. Duke-Cohan, R.E. Hussey, P.W. Tetteh, M. Hilton, G. Wagner, M.J. Lang, and E.L. Reinherz, Pre-T Cell Receptors (Pre-TCRs) Leverage Vbeta Complementarity Determining Regions (CDRs) and Hydrophobic Patch in Mechanosensing Thymic Self-ligands. *J Biol Chem*, 2016. **291**(49):25292-25305.
12. Mallis, R.J., K. Bai, H. Arthanari, R.E. Hussey, M. Handley, Z. Li, L. Chingozha, J.S. Duke-Cohan, H. Lu, J.H. Wang, C. Zhu, G. Wagner, and E.L. Reinherz, Pre-TCR ligand binding impacts thymocyte development before alphabetaTCR expression. *Proc Natl Acad Sci U S A*, 2015. **112**(27):8373-8.
13. Zhao, X., E.M. Kolawole, W. Chan, Y. Feng, X. Yang, M.H. Gee, K.M. Jude, L.V. Sibener, P.M. Fordyce, R.N. Germain, B.D. Evavold, and K.C. Garcia, Tuning T cell receptor sensitivity through catch bond engineering. *Science*, 2022. **376**(6589):eabl5282.
14. Marshall, B.T., M. Long, J.W. Piper, T. Yago, R.P. McEver, and C. Zhu, Direct observation of catch bonds involving cell-adhesion molecules. *Nature*, 2003. **423**(6936):190-3.
15. Sarangapani, K.K., T. Yago, A.G. Klopocki, M.B. Lawrence, C.B. Fieger, S.D. Rosen, R.P. McEver, and C. Zhu, Low force decelerates L-selectin dissociation from P-selectin glycoprotein ligand-1 and endoglycan. *J. Biol. Chem.*, 2004. **279**(3):2291-2298.

16. Wayman, A.M., W. Chen, R.P. McEver, and C. Zhu, Triphasic force dependence of E-selectin/ligand dissociation governs cell rolling under flow. *Biophys. J.*, 2010. **99**(4):1166-74.
17. Choi, Y.I., J.S. Duke-Cohan, W. Chen, B. Liu, J. Rossy, T. Tabarin, L. Ju, J. Gui, K. Gaus, C. Zhu, and E.L. Reinherz, Dynamic control of beta1 integrin adhesion by the plexinD1-sema3E axis. *Proc. Natl. Acad. Sci. U.S.A.*, 2014. **111**(1):379-84.
18. Rosetti, F., Y. Chen, M. Sen, E. Thayer, V. Azcutia, J.M. Herter, F.W. Luscinikas, X. Cullere, C. Zhu, and T.N. Mayadas, A Lupus-Associated Mac-1 Variant Has Defects in Integrin Allosterity and Interaction with Ligands under Force. *Cell Rep*, 2015. **10**(10):1655-1664.
19. Ju, L., Y. Chen, F. Zhou, H. Lu, M.A. Cruz, and C. Zhu, Von Willebrand factor-A1 domain binds platelet glycoprotein Ibalpha in multiple states with distinctive force-dependent dissociation kinetics. *Thromb Res*, 2015. **136**(3):606-12.
20. Yago, T., J. Lou, T. Wu, J. Yang, J.J. Miner, L. Coburn, J.A. Lopez, M.A. Cruz, J.F. Dong, L.V. McIntire, R.P. McEver, and C. Zhu, Platelet glycoprotein Ibalpha forms catch bonds with human WT vWF but not with type 2B von Willebrand disease vWF. *J. Clin. Invest.*, 2008. **118**(9):3195-207.
21. Saggiu, G., K. Okubo, Y. Chen, R. Vattepu, N. Tsuboi, F. Rosetti, X. Cullere, N. Washburn, S. Tahir, A.M. Rosado, S.M. Holland, R.M. Anthony, M. Sen, C. Zhu, and T.N. Mayadas, Cis interaction between sialylated FcgammaRIIA and the alphas-domain of Mac-1 limits antibody-mediated neutrophil recruitment. *Nat Commun*, 2018. **9**(1):5058.
22. Rosado, A.M., Y. Zhang, H. Kyu Choi, S.M. Ehrlich, F. Jin, A. Grakoui, B.D. Evavold, and C. Zhu, Memory in repetitive protein-protein interaction series. *APL Bioengineering*, 2022:in press.
23. Abels, J.A., F. Moreno-Herrero, T. van der Heijden, C. Dekker, and N.H. Dekker, Single-molecule measurements of the persistence length of double-stranded RNA. *Biophys J*, 2005. **88**(4):2737-44.
24. Chen, H., H. Fu, X. Zhu, P. Cong, F. Nakamura, and J. Yan, Improved high-force magnetic tweezers for stretching and refolding of proteins and short DNA. *Biophys J*, 2011. **100**(2):517-23.
25. Choi, H.K., H.G. Kim, M.J. Shon, and T.Y. Yoon, High-Resolution Single-Molecule Magnetic Tweezers. *Annu Rev Biochem*, 2022. **91**:33-59.
26. Feng, Y., K.N. Brazin, E. Kobayashi, R.J. Mallis, E.L. Reinherz, and M.J. Lang, Mechanosensing drives acuity of alphabeta T-cell recognition. *Proc Natl Acad Sci U S A*, 2017. **114**(39):E8204-E8213.
27. Pedersen, L.O., A. Stryhn, T.L. Holter, M. Etzerodt, J. Gerwien, M.H. Nissen, H.C. Thogersen, and S. Buus, The interaction of beta 2-microglobulin (beta 2m) with mouse class I major histocompatibility antigens and its ability to support peptide binding. A comparison of human and mouse beta 2m. *Eur J Immunol*, 1995. **25**(6):1609-16.
28. Benoit, L.A. and R. Tan, Xenogeneic beta 2-microglobulin substitution affects functional binding of MHC class I molecules by CD8+ T cells. *J. Immunol.*, 2007. **179**(6):3588-95.
29. Natarajan, A., V. Nadarajah, K. Felsovalyi, W. Wang, V.R. Jeyachandran, R.A. Wasson, T. Cardozo, C. Bracken, and M. Krogsaard, Structural Model of the Extracellular Assembly of the TCR-CD3 Complex. *Cell Rep*, 2016. **14**(12):2833-45.
30. Dong, L. Zheng, J. Lin, B. Zhang, Y. Zhu, N. Li, S. Xie, Y. Wang, N. Gao, and Z. Huang, Structural basis of assembly of the human T cell receptor-CD3 complex. *Nature*, 2019.
31. Susac, L., M.T. Vuong, C. Thomas, S. von Bulow, C. O'Brien-Ball, A.M. Santos, R.A. Fernandes, G. Hummer, R. Tampe, and S.J. Davis, Structure of a fully assembled tumor-specific T cell receptor ligated by pMHC. *Cell*, 2022. **185**(17):3201-3213 e19.
32. Limozin, L., M. Bridge, P. Bongrand, O. Dushek, P.A. van der Merwe, and P. Robert, TCR-pMHC kinetics under force in a cell-free system show no intrinsic catch bond, but a minimal encounter duration before binding. *Proc Natl Acad Sci U S A*, 2019. **116**(34):16943-16948.
33. Lou, J., T. Yago, A.G. Klopocki, P. Mehta, W. Chen, V.I. Zarnitsyna, N.V. Bovin, C. Zhu, and R.P. McEver, Flow-enhanced adhesion regulated by a selectin interdomain hinge. *J. Cell Biol.*, 2006. **174**(7):1107-17.
34. Chen, Y., L.A. Ju, F. Zhou, J. Liao, L. Xue, Q.P. Su, D. Jin, Y. Yuan, H. Lu, S.P. Jackson, and C. Zhu, An integrin alphaIIb beta3 intermediate affinity state mediates biomechanical platelet aggregation. *Nat Mater*, 2019. **18**(7):760-769.

35. Yago, T., J. Wu, C.D. Wey, A.G. Klopocki, C. Zhu, and R.P. McEver, Catch bonds govern adhesion through L-selectin at threshold shear. *J Cell Biol*, 2004. **166**(6):913-23.
36. Pettmann, Johannes, et al. Mechanical forces impair antigen discrimination by reducing differences in T-cell receptor/peptide–MHC off-rates. *The EMBO Journal* (2022): e2500
37. Gohring, J., F. Kellner, L. Schrangl, R. Platzer, E. Klotzsch, H. Stockinger, J.B. Huppa, and G.J. Schutz, Temporal analysis of T-cell receptor-imposed forces via quantitative single molecule FRET measurements. *Nat Commun*, 2021. **12**(1):2502.
38. Liu, Y., L. Blanchfield, V.P. Ma, R. Andargachew, K. Galior, Z. Liu, B. Evavold, and K. Salaita, DNA-based nanoparticle tension sensors reveal that T-cell receptors transmit defined pN forces to their antigens for enhanced fidelity. *Proc Natl Acad Sci U S A*, 2016. **113**(20):5610-5.
39. Ma, R., A.V. Kellner, V.P. Ma, H. Su, B.R. Deal, J.M. Brockman, and K. Salaita, DNA probes that store mechanical information reveal transient piconewton forces applied by T cells. *Proc Natl Acad Sci U S A*, 2019. **116**(34):16949-16954.
40. Ma, V.P., Y. Liu, L. Blanchfield, H. Su, B.D. Evavold, and K. Salaita, Ratiometric Tension Probes for Mapping Receptor Forces and Clustering at Intermembrane Junctions. *Nano Lett*, 2016. **16**(7):4552-9.
41. Wang, M.S., Y. Hu, E.E. Sanchez, X. Xie, N.H. Roy, M. de Jesus, B.Y. Winer, E.A. Zale, W. Jin, C. Sachar, J.H. Lee, Y. Hong, M. Kim, L.C. Kam, K. Salaita, and M. Huse, Mechanically active integrins target lytic secretion at the immune synapse to facilitate cellular cytotoxicity. *Nat Commun*, 2022. **13**(1):3222.
42. Zhang, Y., C. Ge, C. Zhu, and K. Salaita, DNA-based digital tension probes reveal integrin forces during early cell adhesion. *Nat Commun*, 2014. **5**:5167.
43. Brenner, M.D., R. Zhou, D.E. Conway, L. Lanzano, E. Gratton, M.A. Schwartz, and T. Ha, Spider Silk Peptide Is a Compact, Linear Nanospring Ideal for Intracellular Tension Sensing. *Nano Lett*, 2016. **16**(3):2096-102.
44. Seol, Y., J. Li, P.C. Nelson, T.T. Perkins, and M.D. Betterton, Elasticity of short DNA molecules: theory and experiment for contour lengths of 0.6-7 microm. *Biophys J*, 2007. **93**(12):4360-73.
45. Becker, N., E. Oroudjev, S. Mutz, J.P. Cleveland, P.K. Hansma, C.Y. Hayashi, D.E. Makarov, and H.G. Hansma, Molecular nanosprings in spider capture-silk threads. *Nat Mater*, 2003. **2**(4):278-83.
46. Yanagida, T. and A.H. Iwane, A large step for myosin. *Proc Natl Acad Sci U S A*, 2000. **97**(17):9357-9.
47. Takagi, Y., E.E. Homsher, Y.E. Goldman, and H. Shuman, Force generation in single conventional actomyosin complexes under high dynamic load. *Biophys J*, 2006. **90**(4):1295-307.
48. Ma, R., A.V. Kellner, Y. Hu, B.R. Deal, A.T. Blanchard, and K. Salaita, DNA Tension Probes to Map the Transient Piconewton Receptor Forces by Immune Cells. *J Vis Exp*, 2021(169).
49. Chang, A.C., A.H. Mekhdjian, M. Morimatsu, A.K. Denisin, B.L. Pruitt, and A.R. Dunn, Single Molecule Force Measurements in Living Cells Reveal a Minimally Tensioned Integrin State. *ACS Nano*, 2016. **10**(12):10745-10752.
50. Chowdhury, F., I.T.S. Li, B.J. Leslie, S. Doganay, R. Singh, X. Wang, J. Seong, S.H. Lee, S. Park, N. Wang, and T. Ha, Single molecular force across single integrins dictates cell spreading. *Integr Biol (Camb)*, 2015. **7**(10):1265-1271.
51. Wang, X. and T. Ha, Defining single molecular forces required to activate integrin and notch signaling. *Science*, 2013. **340**(6135):991-4.
52. Rashid, S.A., A.T. Blanchard, J.D. Combs, N. Fernandez, Y. Dong, H.C. Cho, and K. Salaita, DNA Tension Probes Show that Cardiomyocyte Maturation Is Sensitive to the Piconewton Traction Forces Transmitted by Integrins. *ACS Nano*, 2022.
53. Perez, L.A., A. Rashid, J.D. Combs, P. Schneider, A. Rodriguez, K. Salaita, and L. Leyton, An Outside-In Switch in Integrin Signaling Caused by Chemical and Mechanical Signals in Reactive Astrocytes. *Front Cell Dev Biol*, 2021. **9**:712627.

54. Rao, T.C., V.P. Ma, A. Blanchard, T.M. Urner, S. Grandhi, K. Salaita, and A.L. Mattheyses, EGFR activation attenuates the mechanical threshold for integrin tension and focal adhesion formation. *J Cell Sci*, 2020. **133**(13).
55. Brockman, J.M., A.T. Blanchard, V.M. Pui-Yan, W.D. Derricotte, Y. Zhang, M.E. Fay, W.A. Lam, F.A. Evangelista, A.L. Mattheyses, and K. Salaita, Mapping the 3D orientation of piconewton integrin traction forces. *Nat Methods*, 2018. **15**(2):115-118.
56. Glazier, R., J.M. Brockman, E. Bartle, A.L. Mattheyses, O. Destaing, and K. Salaita, DNA mechanotechnology reveals that integrin receptors apply pN forces in podosomes on fluid substrates. *Nat Commun*, 2019. **10**(1):4507.
57. Zhang, Y., Y. Qiu, A.T. Blanchard, Y. Chang, J.M. Brockman, V.P. Ma, W.A. Lam, and K. Salaita, Platelet integrins exhibit anisotropic mechanosensing and harness piconewton forces to mediate platelet aggregation. *Proc Natl Acad Sci U S A*, 2018. **115**(2):325-330.
58. Hui, K.L., L. Balagopalan, L.E. Samelson, and A. Upadhyaya, Cytoskeletal forces during signaling activation in Jurkat T-cells. *Mol Biol Cell*, 2015. **26**(4):685-95.
59. Hui, K.L. and A. Upadhyaya, Dynamic microtubules regulate cellular contractility during T-cell activation. *Proc Natl Acad Sci U S A*, 2017. **114**(21):E4175-E4183.
60. Bashour, K.T., A. Gondarenko, H. Chen, K. Shen, X. Liu, M. Huse, J.C. Hone, and L.C. Kam, CD28 and CD3 have complementary roles in T-cell traction forces. *Proc. Natl. Acad. Sci. U.S.A.*, 2014. **111**(6):2241-6.
61. Bashour, K.T., J. Tsai, K. Shen, J.H. Lee, E. Sun, M.C. Milone, M.L. Dustin, and L.C. Kam, Cross talk between CD3 and CD28 is spatially modulated by protein lateral mobility. *Mol. Cell. Biol.*, 2014. **34**(6):955-64.
62. Munevar, S., Y. Wang, and M. Dembo, Traction force microscopy of migrating normal and H-ras transformed 3T3 fibroblasts. *Biophys J*, 2001. **80**(4):1744-57.
63. Tan, J.L., J. Tien, D.M. Pirone, D.S. Gray, K. Bhadriraju, and C.S. Chen, Cells lying on a bed of microneedles: an approach to isolate mechanical force. *Proc Natl Acad Sci U S A*, 2003. **100**(4):1484-9.
64. Klotzsch, E. and G.J. Schutz, Improved ligand discrimination by force-induced unbinding of the T cell receptor from peptide-MHC. *Biophys. J.*, 2013. **104**(8):1670-5.
65. Pereverzev, Y.V., O.V. Prezhdo, M. Forero, E.V. Sokurenko, and W.E. Thomas, The two-pathway model for the catch-slip transition in biological adhesion. *Biophys J*, 2005. **89**(3):1446-54.

REVIEWERS' COMMENTS

Reviewer #1 (Remarks to the Author):

In their revised version Choi et al. have addressed a number of my previous concerns/suggestions. The wording has been modified in some instances to provide a more balanced view of the current state of progress. As such, the ms. has improved in readability, both in terms of supporting graphics and the provision / accessibility of data as well as experimental parameters. The authors have also included more work, in response to other reviewers of the ms., which strengthens their claims.

While I have not changed my overall position with regard to many of the issues I have previously raised, I acknowledge the toned-down/ more nuanced wording and the additional experimental investment which - together with the to be published correspondence between authors and reviewers - strengthen the main message of this ms.. I am convinced that in its current form, the work by Choi et al. will be thoroughly studied both inside and outside the TCR-force field.

Most importantly, the presented data create sufficient momentum to challenge current perceptions and to result in testable hypotheses (e.g. drastic structural changes within MHC class I upon TCR-engagement). The verdict is still out and future experiments will tell, which is absolutely in line with the scientific method. I hence welcome publication of the revised version in Nat. Comm., especially if points 2 and 3 (below) are satisfactorily addressed.

Minor points:

1. I appreciate the data shown in Fig. R1 (of the rebuttal), but they address the concerns in the field only in part. It would be telling to begin the BFP experiments at high force (e.g. 15 to 20 pN) where longest bond lifetimes are typically observed, and then dial down the forces gradually to 0pN or, alternatively, dial them up to 28pN. Another approach would be to pick individual T cells for individual force regimens applied: i.e. measure X cells at 0pN, Y cells (different from X) at 5 pN, and so on ... to avoid any history of previous force encounters.

2. In their 2023 paper, Pettman et al. (PMID: 36484367) observe with the use of laminar flow catch bonds for the OT1 system (which features unusually short-lived agonist-TCR interactions), yet see antigen discrimination improved at low force (in essence lower affinity interactions appear to suffer less from applied forces than higher affinity interactions). This message goes against conclusions of this ms., and therefore the Pettmann paper should be discussed.

3. The authors may want to cite (ideally in their introduction) evidence for BFP-single molecule sensitivity for TCR-pMHC interactions, in particular for low affinity TCR-pMHC interactions. If this is not possible, a convincing explanation could build trust.

Reviewer #2 (Remarks to the Author):

This reviewer commends the authors for the extensive and comprehensive response the Reviewers' comments. The authors have adequately addressed my concerns and I note that the manuscript is improved and more accessible. - Paolo Provenzano

Reviewer #3 (Remarks to the Author):

I am happy with the changes in response to my comments.

RESPONSE TO REVIEWER COMMENTS

Reviewer #1:

In their revised version Choi et al. have addressed a number of my previous concerns/suggestions. The wording has been modified in some instances to provide a more balanced view of the current state of progress. As such, the ms. has improved in readability, both in terms of supporting graphics and the provision / accessibility of data as well as experimental parameters. The authors have also included more work, in response to other reviewers of the ms., which strengthens their claims.

We thank the Reviewer for the positive comments.

While I have not changed my overall position with regard to many of the issues I have previously raised, I acknowledge the toned-down/ more nuanced wording and the additional experimental investment which - together with the to be published correspondence between authors and reviewers - strengthen the main message of this ms.. I am convinced that in its current form, the work by Choi et al. will be thoroughly studied both inside and outside the TCR-force field.

We agree to disagree, and thank the Reviewer for acknowledging the improvement of the revised ms..

Most importantly, the presented data create sufficient momentum to challenge current perceptions and to result in testable hypotheses (e.g. drastic structural changes within MHC class I upon TCR-engagement). The verdict is still out and future experiments will tell, which is absolutely in line with the scientific method. I hence welcome publication of the revised version in Nat. Comm., especially if points 2 and 3 (below) are satisfactorily addressed.

We thank the Reviewer for recommending the publication of our ms..

Minor points:

1. I appreciate the data shown in Fig. R1 (of the rebuttal), but they address the concerns in the field only in part. It would be telling to begin the BFP experiments at high force (e.g. 15 to 20 pN) where longest bond lifetimes are typically observed, and then dial down the forces gradually to 0pN or, alternatively, dial them up to 28pN. Another approach would be to pick individual T cells for individual force regimens applied: i.e. measure X cells at 0pN, Y cells (different from X) at 5 pN, and so on ... to avoid any history of previous force encounters.

If we understand correctly, the type of experiments suggested the Reviewer are called cyclic mechanical reinforcement (CMR), which may be related to catch bond in some way because it greatly prolongs bond lifetime, but it is definitely distinct from catch bond. We published the CMR experiment first in 2013 for integrin $\alpha_5\beta_1$ interaction with fibronectin using AFM and BFP [1]. The data suggest that CMR has more to do with force-induced protein conformational change rather than with cell activation because CMR prolonged bond lifetime much more with purified ectodomain $\alpha_5\beta_1$ protein and native integrin expressed on live cell surface [1]. In a more recently in 2019 paper we showed CMR in G-actin–G-actin and G-actin–F-actin interactions using AFM [2]. There was no live cell in that study; only purified proteins were used. The mechanism underlying CMR is not fully understood but is related to a more general phenomenon we termed force history-dependence, which we published as early as 2005 on P-selectin [3], 2011 on L-selectin [4], and more recently in 2019 on GPIIb α , integrin $\alpha_{IIb}\beta_3$, and TCR using BFP [5]. We know via personal communication that CMR has also been observed in TCR–pMHC interaction using live T cells. Because this privileged

information was provided to us in confidence, we are not at the liberty to disclose these unpublished results. But we wish to re-iterate that the experiment suggested by the Reviewer would not provide the results for which he/she wished. Instead, it will bring in a new level of complication beyond the scope of the present paper.

2. In their 2023 paper, Pettman et al. (PMID: 36484367) observe with the use of laminar flow catch bonds for the OT1 system (which features unusually short-lived agonist-TCR interactions), yet see antigen discrimination improved at low force (in essence lower affinity interactions appear to suffer less from applied forces than higher affinity interactions). This message goes against conclusions of this ms., and therefore the Pettmann paper should be discussed.

In our Response to Reviewer after the first round of review, we mentioned the Pettman et al. paper when it was posted in biorxiv as a preprint. Now that it is published we have cited and discussed it in our paper.

3. The authors may want to cite (ideally in their introduction) evidence for BFP-single molecule sensitivity for TCR-pMHC interactions, in particular for low affinity TCR-pMHC interactions. If this is not possible, a convincing explanation could build trust.

In the revised manuscript, we have cited the original paper of Evans et al. in 1995 reporting the invention of BFP [6] and two papers we published in 2008 and 2017, which specifically address the technical specifications and suitability of BFP for single-molecule experiment [7, 8].

Reviewer #2:

This reviewer commends the authors for the extensive and comprehensive response the Reviewers' comments. The authors have adequately addressed my concerns and I note that the manuscript is improved and more accessible. - Paolo Provenzano

We thank the Reviewer for his kind words and endorsement.

Reviewer #3:

I am happy with the changes in response to my comments.

We thank the Reviewer for his/her satisfaction of our response to his/her comments.

References

1. Kong, F., Z. Li, W.M. Parks, D.W. Dumbauld, A.J. Garcia, A.P. Mould, M.J. Humphries, and C. Zhu, Cyclic mechanical reinforcement of integrin-ligand interactions. *Mol Cell*, 2013. **49**(6):1060-8.
2. Lee, H., S.G. Eskin, S. Ono, C. Zhu, and L.V. McIntire, Force-history dependence and cyclic mechanical reinforcement of actin filaments at the single molecular level. *J Cell Sci*, 2019. **132**(4).
3. Marshall, B.T., K.K. Sarangapani, J. Lou, R.P. McEver, and C. Zhu, Force history dependence of receptor-ligand dissociation. *Biophys J*, 2005. **88**(2):1458-66.
4. Sarangapani, K.K., J. Qian, W. Chen, V.I. Zarnitsyna, P. Mehta, T. Yago, R.P. McEver, and C. Zhu, Regulation of catch bonds by rate of force application. *J Biol Chem*, 2011. **286**(37):32749-61.
5. Chen, Y., J. Liao, Y. Zhou, K. Li, B. Liu, L.A. Ju, and C. Zhu, Fast force loading disrupts molecular bond stability in human and mouse cell adhesions. *Molecular & Cellular Biomechanics*, 2019. **16**:97.
6. Evans, E., K. Ritchie, and R. Merkel, Sensitive force technique to probe molecular adhesion and structural linkages at biological interfaces. *Biophys J*, 1995. **68**(6):2580-7.
7. Chen, W., V.I. Zarnitsyna, K.K. Sarangapani, J. Huang, and C. Zhu, Measuring receptor-ligand binding kinetics on cell surfaces: From adhesion frequency to thermal fluctuation methods. *Cell Mol. Bioeng.*, 2008. **1**(4):276-288.
8. Ju, L. and C. Zhu, Benchmarks of Biomembrane Force Probe Spring Constant Models. *Biophys J*, 2017. **113**(12):2842-2845.